# FAST REGRESSION FOR STRUCTURED INPUTS

**Raphael A. Meyer**
New York University
ram900@nyu.edu

**Cameron Musco**
University of Massachusetts Amherst
cmusco@cs.umass.edu

**Christopher Musco**
New York University
cmusco@nyu.edu

**David P. Woodruff**
Carnegie Mellon University
dwoodruf@andrew.cmu.edu

**Samson Zhou**
Carnegie Mellon University
samsonzhou@gmail.com

## ABSTRACT

We study the $\ell_p$ regression problem, which requires finding $\mathbf{x} \in \mathbb{R}^d$ that minimizes $\|\mathbf{Ax} - \mathbf{b}\|_p$ for a matrix $\mathbf{A} \in \mathbb{R}^{n \times d}$ and response vector $\mathbf{b} \in \mathbb{R}^n$. There has been recent interest in developing subsampling methods for this problem that can outperform standard techniques when $n$ is very large. However, all known subsampling approaches have run time that depends exponentially on $p$, typically, $d^{\mathcal{O}(p)}$, which can be prohibitively expensive. We improve on this work by showing that for a large class of common *structured matrices*, such as combinations of low-rank matrices, sparse matrices, and Vandermonde matrices, there are subsampling based methods for $\ell_p$ regression that depend polynomially on $p$. For example, we give an algorithm for $\ell_p$ regression on Vandermonde matrices that runs in time $\mathcal{O}(n \log^3 n + (dp^2)^{0.5+\omega} \cdot \text{polylog}\, n)$, where $\omega$ is the exponent of matrix multiplication. The polynomial dependence on $p$ crucially allows our algorithms to extend naturally to efficient algorithms for $\ell_\infty$ regression, via approximation of $\ell_\infty$ by $\ell_{\mathcal{O}(\log n)}$. Of practical interest, we also develop a new subsampling algorithm for $\ell_p$ regression for arbitrary matrices, which is simpler than previous approaches for $p \geq 4$.

## 1 INTRODUCTION

Given a matrix $\mathbf{A} \in \mathbb{R}^{n \times d}$ and a vector $\mathbf{b} \in \mathbb{R}^n$, the goal of linear regression is to find a vector $\mathbf{x} \in \mathbb{R}^d$ such that $\mathbf{Ax}$ is as close as possible to $\mathbf{b}$. In approximate $\ell_p$ linear regression in particular, we seek to find $\tilde{\mathbf{x}} \in \mathbb{R}^d$ such that, for some approximation parameter $\varepsilon > 0$,

$$\|\mathbf{A\tilde{x}} - \mathbf{b}\|_p \leq (1 + \varepsilon) \min_{\mathbf{x} \in \mathbb{R}^d} \|\mathbf{Ax} - \mathbf{b}\|_p.$$

Here for a vector $\mathbf{y} \in \mathbb{R}^n$, $\|\mathbf{y}\|_p = \left(\sum_{i=1}^n |\mathbf{y}_i|^p\right)^{1/p}$. $\ell_p$ regression is central in statistical data analysis, and has numerous applications in machine learning, data science, and applied mathematics (Friedman et al., 2001; Chatterjee & Hadi, 2006). There are a number of algorithmic approaches to solving $\ell_p$ regression. For example, we can directly apply iterative methods like gradient descent or stochastic gradient descent. Alternatively, we can use iteratively reweighted least squares, which reduces the regression problem to solving $\text{poly}(d)$ linear systems (Adil et al., 2019b;a).

Both the above approaches require repeated passes over the matrix $\mathbf{A}$, so while their runtimes are typically linear in $nd$, or more generally on the time to multiply the matrix $\mathbf{A}$ by a vector, that factor is multiplied by other parameters, such as the number of iterations to convergence. An alternative approach, which can lead to faster running time when $n$ is large, is to apply "sketch-and-solve" methods. This approaches begins with an inexpensive subsampling step, which selects a subset of rows in $\mathbf{A}$ to produce a smaller matrix $\mathbf{M}$ with $\text{poly}\left(d, \frac{1}{\varepsilon}, \log n\right) \ll n$ rows. $\mathbf{M}$ can be written as $\mathbf{M} = \mathbf{SA}$ where $\mathbf{S}$ is a row sampling and rescaling matrix. The goal is for $\|\mathbf{SAx} - \mathbf{Sb}\|_p$ to be a good approximation to $\|\mathbf{Ax} - \mathbf{b}\|_p$ for all $\mathbf{x} \in \mathbb{R}^d$. If this the case, then an approximate solution to the original $\ell_p$ regression problem can be obtained by solving the subsampled problem, which has smaller size, thus allowing for more efficient computation.

The standard approach to subsampling for $\ell_p$ regression is to sample rows with probability proportional to their so-called $\ell_p$ *Lewis weights* (Cohen & Peng, 2015). Unfortunately, for general inputs $\mathbf{A}$, $\ell_p$ Lewis weight sampling requires $\mathcal{O}\left(d^{\max(1,p/2)}\right)$ rows, and it can be shown that no subsampling method can take fewer than $d^{\mathcal{O}(p)}$ (Li et al., 2021). This means that sampling is only helpful in the limited regime where $n \gg d^{p/2}$. However, there are many applications in which the matrix $\mathbf{A}$ has additional structure, which can be leveraged to design more efficient algorithms. For example, Vandermonde matrices are used in the polynomial regression problem, which has been studied for over 200 years (Gergonne, 1974) and has applications to machine learning (Kalai et al., 2008), applied statistics (MacNeill, 1978), and computer graphics (Pratt, 1987). The goal is to fit a signal, which is measured at time points $t_1, \ldots, t_n$ using a degree $d$ polynomial. This problem can be formulated as $\ell_p$ regression with a Vandermonde feature matrix $\mathbf{A}$, whose $i^{th}$ row $\mathbf{a}_i$ is of the form $[1, t_i, (t_i)^2, \ldots, (t_i)^{d-1}]$. Regression problems with Vandermonde matrices also arise in the settings of Fourier-constrained function fitting (Avron et al., 2019) and Toeplitz covariance estimation (Eldar et al., 2020). Notably, (Shi & Woodruff, 2019) leverages the structure of Vandermonde matrices to more quickly build a subsampled matrix $\mathbf{M}$ given any Vandermonde matrix $\mathbf{A}$ than would be possible for a general input. Their method does not change how many rows are in the matrix $\mathbf{M}$, so the overall algorithm still incurs an exponential dependence in $p$. So while the approach is a helpful improvement for Vandermonde regression where $p$ is small, this leaves an infeasible runtime for important problems like $\ell_\infty$ regression, which can be approximated by $\ell_p$ regression for $p = \mathcal{O}(\log n)$.

## 1.1 OUR CONTRIBUTIONS

We first show that for $\ell_p$ regression on Vandermonde matrices, it is possible to reduce the size of the subsampled matrix $\mathbf{M}$ to depend polynomially instead of exponentially on $p$.

**Theorem 1.1.** *Given $\varepsilon \in (0, 1)$ and $p \geq 1$, a Vandermonde matrix $\mathbf{A} \in \mathbb{R}^{n \times d}$, and $\mathbf{b} \in \mathbb{R}^n$, there exists an algorithm that uses $\mathcal{O}\left(n \log^3 n\right) + d^{0.5+\omega} \operatorname{poly}\left(\frac{1}{\varepsilon}, p, \log n\right)$ time to compute a sampling matrix $\mathbf{S} \in \mathbb{R}^{m \times n}$ with $m = \mathcal{O}\left(\frac{p^2 d}{\varepsilon^3} \log^2 n\right)$ rows so that with high probability, for all $\mathbf{x} \in \mathbb{R}^d$,*

$$(1 - \varepsilon)\|\mathbf{A}\mathbf{x} - \mathbf{b}\|_p \leq \|\mathbf{S}\mathbf{A}\mathbf{x} - \mathbf{S}\mathbf{b}\|_p \leq (1 + \varepsilon)\|\mathbf{A}\mathbf{x} - \mathbf{b}\|_p,$$

*and then to return a vector $\widehat{\mathbf{x}} \in \mathbb{R}^d$ such that $\|\mathbf{A}\widehat{\mathbf{x}} - \mathbf{b}\|_p \leq (1 + \varepsilon) \min_{\mathbf{x} \in \mathbb{R}^d} \|\mathbf{A}\mathbf{x} - \mathbf{b}\|_p$.*

The best known previous work required at least $\Omega(n \log^2 n + d^{p/2} \cdot \operatorname{poly}(1/\varepsilon))$ time (Shi & Woodruff, 2019). This has an exponential dependence in $p$, while our algorithm has just a polynomial dependence.

Building on Theorem 1.1, we observe that to obtain a $(1 + \varepsilon)$-approximation to the fundamental problem of $\ell_\infty$ polynomial regression, it suffices to consider $\ell_p$ regression for $p = \mathcal{O}\left(\frac{\log n}{\varepsilon}\right)$. Since our results have polynomial dependence on $p$ rather than exponential, we thus obtain the first subsampling guarantees for Vandermonde $\ell_\infty$ regression.

**Theorem 1.2.** *Given $\varepsilon \in (0, 1)$, a Vandermonde matrix $\mathbf{A} \in \mathbb{R}^{n \times d}$, and $\mathbf{b} \in \mathbb{R}^d$, there exists an algorithm that uses $\mathcal{O}\left(n \log^3 n\right) + d^{0.5+\omega} \operatorname{poly}\left(\frac{1}{\varepsilon}, \log n\right)$ time to compute a sampling matrix $\mathbf{S} \in \mathbb{R}^{m \times n}$ with $m = \mathcal{O}\left(\frac{d}{\varepsilon^5} \log^4 n\right)$ such that with high probability, for all $\mathbf{x} \in \mathbb{R}^d$,*

$$(1 - \varepsilon)\|\mathbf{A}\mathbf{x} - \mathbf{b}\|_\infty \leq \|\mathbf{S}\mathbf{A}\mathbf{x} - \mathbf{S}\mathbf{b}\|_\infty \leq (1 + \varepsilon)\|\mathbf{A}\mathbf{x} - \mathbf{b}\|_\infty,$$

*and then to return a vector $\widehat{\mathbf{x}} \in \mathbb{R}^d$ such that $\|\mathbf{A}\widehat{\mathbf{x}} - \mathbf{b}\|_\infty \leq (1 + \varepsilon) \min_{\mathbf{x} \in \mathbb{R}^d} \|\mathbf{A}\mathbf{x} - \mathbf{b}\|_\infty$.*

To the best of our knowledge, this is the first known dimensionality reduction for $\ell_\infty$ regression with provable guarantees for any (nontrivial) input matrix. We summarize these results in Table 1.

Our second contribution is to show that improved sampling bounds for $\ell_p$ regression can be extended to a broad class of inputs, beyond Vandemonde matrices. We introduce the following definition to capture the "true dimension" of regression problems for structured input matrices.

**Definition 1.3** (Rank of Regression Problem)**.** *Given an integer $p \geq 1$ and a matrix $\mathbf{A} \in \mathbb{R}^{n \times d}$, suppose there exists a matrix $\mathbf{M} \in \mathbb{R}^{n \times t}$ and a fixed function $f : \mathbb{R}^d \to \mathbb{R}^t$ so that for all $\mathbf{x} \in \mathbb{R}^d$,*

$$|\langle \mathbf{a}_i, \mathbf{x} \rangle|^p = |\langle \mathbf{m}_i, f(\mathbf{x}) \rangle|.$$

*Then we call the minimal such $t$ the* rank *of the $\ell_p$ regression problem.*

| Rows Sampled, $\ell_p$ Regression | Rows Sampled, $\ell_\infty$ Regression | Reference |
|---|---|---|
| $d^p \operatorname{poly}\left(\log n, \frac{1}{\varepsilon}\right)$ | $n$ | (Avron et al., 2013) |
| $d^p \operatorname{poly}\left(\log n, \frac{1}{\varepsilon}\right)$ | $n$ | (Shi & Woodruff, 2019) |
| $dp^2 \operatorname{poly}\left(\log n, \frac{1}{\varepsilon}\right)$ (Theorem 1.1) | $d \operatorname{poly}\left(\log n, \frac{1}{\varepsilon}\right)$ (Theorem 1.2) | Our Results |

Table 1: Sample complexity for regression on Vandermonde matrices

Theorems 1.1 and 1.2 rely on the following key structural property that we prove about the $p$-fold tensor product of rows of the Vandermonde matrix.

**Lemma 1.4.** *For integer $p \geq 1$, the rank of the $\ell_p$ regression problem on a Vandermonde matrix $\mathbf{A} \in \mathbb{R}^{n \times d}$ is $\mathcal{O}(dp)$.*

Lemma 1.4 implies that the $\ell_p$ loss function for a row of a Vandermonde matrix can potentially be expressed as a linear combination of $\mathcal{O}(dp)$ variables even though the entries of the measurement vector $\mathbf{b}$ can be arbitrary. By comparison, the $\ell_p$ loss function for a row $\mathbf{a}_i$ of a general matrix $\mathbf{A}$ is $|\langle \mathbf{a}_i, \mathbf{x} \rangle - b_i|_p^p$ can only be expressed as a linear combination of $\mathcal{O}(pd^p)$ variables, corresponding to each of the $d^k$ $k$-wise products of coordinates of $\mathbf{x}$, for each $k \in [p]$. As a corollary of Lemma 1.4, Theorems 1.1 and 1.2 obtain a small coreset for $\ell_p$ regression (as well as $\ell_\infty$ regression) on a Vandermonde matrix, which can thus be used as a preconditioner for $\ell_p$ regression.

We generalize Lemma 1.4 to similar guarantees for $\ell_p$ regression on a matrix $\mathbf{A}$ that is the sum of a low-rank matrix $\mathbf{K}$ and an $s$-sparse matrix $\mathbf{S}$ that has at most $s$ non-zero entries per row. Thus using the notion of the rank of the regression problem for such a matrix $\mathbf{A}$, we obtain the following guarantee:

**Theorem 1.5.** *Given $\varepsilon \in (0,1)$ and $p \geq 1$, a rank $k$ matrix $\mathbf{K} \in \mathbb{R}^{n \times d}$ and a $s$-sparse matrix $\mathbf{S} \in \mathbb{R}^{n \times d}$ so that $\mathbf{A} := \mathbf{K} + \mathbf{S}$, and $\mathbf{b} \in \mathbb{R}^n$, there exists an algorithm that uses $\mathcal{O}\left(nk^{\omega-1}\right) + n \operatorname{poly}\left(2^p, d^s, k^p, s^p, \frac{1}{\varepsilon}, \log n\right)$ time to compute a sampling matrix $\mathbf{T} \in \mathbb{R}^{m \times n}$ containing $m = \mathcal{O}\left(\frac{pd^s(k+s)^p(s+p)}{\varepsilon^3} \log^2(pn)\right)$ rows so that with high probability, for all $\mathbf{x} \in \mathbb{R}^d$,*

$$(1-\varepsilon)\|\mathbf{A}\mathbf{x} - \mathbf{b}\|_p \leq \|\mathbf{T}\mathbf{A}\mathbf{x} - \mathbf{T}\mathbf{b}\|_p \leq (1+\varepsilon)\|\mathbf{A}\mathbf{x} - \mathbf{b}\|_p,$$

*and then to return a vector $\widehat{\mathbf{x}} \in \mathbb{R}^d$ such that $\|\mathbf{A}\widehat{\mathbf{x}} - \mathbf{b}\|_p \leq (1+\varepsilon) \min_{\mathbf{x} \in \mathbb{R}^d} \|\mathbf{A}\mathbf{x} - \mathbf{b}\|_p$. Further, if the low-rank factorization of $\mathbf{K}$ is given explicitly, then this runtime can be improved to $n \operatorname{poly}\left(2^p, d^s, k^p, s^p, \frac{1}{\varepsilon}, \log n\right)$.*

Similarly, we obtain efficient guarantees for $\ell_p$ regression on a matrix $\mathbf{A}$ that is the sum of a Vandermonde matrix $\mathbf{V}$ and a sparse matrix $\mathbf{S}$ that has at most $s$ non-zero entries per row.

**Theorem 1.6.** *Given $\varepsilon \in (0,1)$ and $p \geq 1$, a Vandermonde matrix $\mathbf{V} \in \mathbb{R}^{n \times d}$ and an $s$-sparse matrix $\mathbf{S} \in \mathbb{R}^{n \times d}$ such that $\mathbf{A} := \mathbf{V} + \mathbf{S}$, and $\mathbf{b} \in \mathbb{R}^n$, there exists an algorithm that uses $\mathcal{O}\left(n \log^3 n + \operatorname{poly}\left(p^2 d, d^s, s^p, \log n, \frac{1}{\varepsilon}\right)\right)$ time to compute a sampling matrix $\mathbf{T} \in \mathbb{R}^{m \times n}$ with $m = \mathcal{O}\left(\frac{p^2 d^{s+1} s^p(s+p)}{\varepsilon^3} \log^2(pn)\right)$ rows, so that with high probability for all $\mathbf{x} \in \mathbb{R}^d$,*

$$(1-\varepsilon)\|\mathbf{A}\mathbf{x} - \mathbf{b}\|_p \leq \|\mathbf{T}\mathbf{A}\mathbf{x} - \mathbf{T}\mathbf{b}\|_p \leq (1+\varepsilon)\|\mathbf{A}\mathbf{x} - \mathbf{b}\|_p,$$

*and then to return a vector $\widehat{\mathbf{x}} \in \mathbb{R}^d$ such that $\|\mathbf{A}\widehat{\mathbf{x}} - \mathbf{b}\|_p \leq (1+\varepsilon) \min_{\mathbf{x} \in \mathbb{R}^d} \|\mathbf{A}\mathbf{x} - \mathbf{b}\|_p$.*

Surprisingly, our methods even yield practical algorithms for general matrices *with absolutely no structure*. Although the rank of the $\ell_p$ regression problem for general matrices is $\mathcal{O}(d^p)$, we still obtain the optimal sample complexity (see (Li et al., 2021) for a lower bound) of roughly $\mathcal{O}\left(d^{p/2}\right)$. Furthermore, an advantage of our approach is that we only need to perform $\ell_q$ Lewis weight sampling for $q \in [1,4]$ and there are known efficient iterative methods for computing the $\ell_q$ Lewis weights for $q \in [1,4]$, e.g., see Section 2.1 of this paper or Section 3 in (Cohen & Peng, 2015). By contrast, previous methods relied on computing general $\ell_p$ weights for $p > 4$, which, prior to the recent work of (Fazel et al., 2021), required solving a large convex program, e.g., through semidefinite programming, see Section 4 in (Cohen & Peng, 2015).

Finally, we experimentally validate our theory on synthetic data. In particular, we consider matrices and response vectors that are motivated by existing lower bounds for subsampling and sketching methods. For Vandermonde regression, our experiments demonstrate that the number of rows

needed is polynomial in $p$, reinforcing how structured matrices outperform the worst-case bound. For unstructured matrix $\ell_p$ regression, we demonstrate that our $\ell_q$ Lewis Weight subsampling scheme is effective and accurate.

## 1.2 OVERVIEW OF OUR TECHNIQUES

Our algorithmic contributions rely on two key observations, which we describe below. For the sake of presentation, we assume $p$ is an integer in this overview, though we handle arbitrary real values of $p$ in the subsequent algorithms and analyses.

**Reduced rank of the $\ell_p$ regression problem on structured inputs.** The first main ingredient is the simple yet powerful observation that the rank of the $\ell_p$ regression problem on structured inputs such as Vandermonde matrices $\mathbf{A} \in \mathbb{R}^{n \times d}$ does not need to be the $\mathcal{O}(d^p)$ rank of the $\ell_p$ regression problem on general matrices. For a general matrix, $\langle \mathbf{a}_i, \mathbf{x} \rangle^p$ for an integer $p$ can be rewritten as a linear combination $\sum \alpha_i y_i$ of $\mathcal{O}(d^p)$ terms, where each coefficient $\alpha_i$ is a $p$-wise product of entries of the row vector $\mathbf{a}_i \in \mathbb{R}^d$ and similarly each $y_i$ is a $p$-wise product of entries of the vector $\mathbf{x} \in \mathbb{R}^d$. However, when $\mathbf{A}$ is a Vandermonde matrix, then the $j$-th entry of $\mathbf{a}_i$ is simply $A_{i,2}^{j-1}$. Thus each coefficient in $\langle \mathbf{a}_i, \mathbf{x} \rangle^p$ can be written as a linear combination of $1, A_{i,2}, (A_{i,2})^2, \ldots, (A_{i,2})^{p(d-1)}$ and so we can express $\langle \mathbf{a}_i, \mathbf{x} \rangle^p$ as a linear combination of $\mathcal{O}(dp)$ variables rather than $\mathcal{O}(d^p)$ variables.

We can similarly show that the rank of the $\ell_p$ regression problem on rank $k$-matrix $\mathbf{A}$ is $\mathcal{O}(k^p)$ by writing each row $\mathbf{a}_i$ as a linear combination of basis vectors $\mathbf{v}_1, \ldots, \mathbf{v}_k$. Then, using the Hadamard Product-Kronecker Product mixed-product property, we can rewrite $\langle \mathbf{a}_i, \mathbf{x} \rangle^p$ as a linear combination of $p$-wise products of the variables $\langle \mathbf{v}_1, \mathbf{x} \rangle, \ldots, \langle \mathbf{v}_k, \mathbf{x} \rangle$, i.e., a linear combination of $\mathcal{O}(k^p)$ variables rather than $\mathcal{O}(d^p)$ variables. It follows that the rank of the $\ell_p$ regression problem on a matrix $\mathbf{A}$ whose rows have at most $s$ non-zero entries is $\mathcal{O}(d^s s^p)$ by noting that (1) there are $\binom{d}{s} = \mathcal{O}(d^s)$ sparsity patterns and that (2) for a fixed sparsity pattern, the rank of the induced matrix is at most $s$, which induces a linear combination of $\mathcal{O}(s^p)$ variables for the $\ell_p$ regression problem. These decomposition techniques can also be generalized to show that the rank of the $\ell_p$ regression problem is low on matrices $\mathbf{A}$ such that $\mathbf{A} = \mathbf{K} + \mathbf{V}$, $\mathbf{A} = \mathbf{K} + \mathbf{S}$ or $\mathbf{A} = \mathbf{V} + \mathbf{S}$, where $\mathbf{K}$ is a low-rank matrix, $\mathbf{V}$ is a Vandermonde matrix, and $\mathbf{S}$ is a sparse matrix.

However, we remark that although the observation that the rank of the $\ell_p$ regression problem on structured inputs can be low, this itself does not yield an algorithm for $\ell_p$ regression. This is because the loss function is $|\langle \mathbf{a}_i, \mathbf{x} \rangle - b_i|^p$ rather than $\langle \mathbf{a}_i, \mathbf{x} \rangle^p$ and there can be $n$ possible different values of $b_i$ across all $i \in [n]$.

**Rounding and truncating the measurement vector: a novel algorithmic technique.** Thus, the second main ingredient is manipulating the measurement vector $\mathbf{b} \in \mathbb{R}^n$ so that the loss function can utilize the low-rank property of the $\ell_p$ regression problem on structured inputs. A natural approach would be to round the entries of $\mathbf{b}$, say to the nearest power of $(1 + \varepsilon)$. Unfortunately, such a rounding approach would roughly preserve $\|\mathbf{b}\|_p$ up to a multiplicative $(1 + \varepsilon)$ factor, but it would not preserve $\|\mathbf{Ax} - \mathbf{b}\|_p$. For example, suppose $\langle \mathbf{a}_i, \mathbf{x} \rangle = b_i = N$ for some arbitrarily large value $N$. Then $|\langle \mathbf{a}_i, \mathbf{x} \rangle - b_i|^p = 0$, but if $b_i$ were rounded to $(1 + \varepsilon)N$, we would have $|\langle \mathbf{a}_i, \mathbf{x} \rangle - b_i|^p = \varepsilon^p N^p$, which can be arbitrarily large.

The lesson from this counterexample is that when $b_i$ is significantly larger than $\langle \mathbf{a}_i, \mathbf{x} \rangle - b_i$, any rounding technique can be arbitrarily bad because $\|\mathbf{b}\|_p$ can be significantly larger than OPT $:= \min_{\mathbf{x} \in \mathbb{R}^d} \|\mathbf{Ax} - \mathbf{b}\|_p$. On the other hand, if $\|\mathbf{b}\|_p$ is a constant factor approximation to OPT, then the previous counterexample cannot happen because either (1) $b_i$ is large relative to $\|\mathbf{b}\|_p$ and $\langle \mathbf{a}_i, \mathbf{x} \rangle$ cannot be too close to $b_i$ so that rounding $b_i$ will not significantly affect the difference $\langle \mathbf{a}_i, \mathbf{x} \rangle - b_i$ or (2) $b_i$ is small relative to $\|\mathbf{b}\|_p$ and so any rounding of $b_i$ will not affect the contribution of the $i$-th row of $\mathbf{A}$ in the overall loss. Thus our task is reduced to manipulating the input so that $\|\mathbf{b}\|_p = \mathcal{O}(\text{OPT})$.

To that end, we note that if $\|\mathbf{A\tilde{x}} - \mathbf{b}\|_p = \mathcal{O}(\text{OPT})$ for a vector $\tilde{\mathbf{x}} \in \mathbb{R}^d$, then by a triangle inequality argument, we have that the residual vector $\mathbf{b}' = \mathbf{b} - \mathbf{A\tilde{x}} \in \mathbb{R}^d$ satisfies $\|\mathbf{b}'\|_p = \mathcal{O}(\text{OPT})$. Namely, we show that to find such a vector $\tilde{\mathbf{x}}$, it suffices by triangle inequality to find a subspace embedding for $\mathbf{A}$, i.e., a matrix $\mathbf{M} \in \mathbb{R}^{m \times d}$ with $m \ll n$ such that

$$(1 - C)\|\mathbf{Ax}\|_p \leq \|\mathbf{Mx}\|_p \leq (1 + C)\|\mathbf{Ax}\|_p$$

for some constant $C \in (0, 1)$. Typically, such a matrix $\mathbf{M}$ can be found by sampling the rows of $\mathbf{A}$ according to their $\ell_p$ Lewis weights to generate a matrix with $m = \mathcal{O}\left(d^{\max(1, p/2)}\right)$ rows. However, using our observation that the rank of the $\ell_p$ regression problem on structured inputs can be low, we can sample $\mathcal{O}(dp)$ rows of $\mathbf{A}$ with probabilities proportional to their $\ell_q$ Lewis weights, where $q = \frac{p}{2^r}$ for an integer $r$ such that $2^r \leq p < 2^{r+1}$. Crucially, we can find such a vector $\tilde{\mathbf{x}}$ *without reading the coordinates* of $\mathbf{b}$ since we only require to read the rows of $\mathbf{A}$ to perform Lewis weight sampling in this phase. Thus we can efficiently find a residual vector $\mathbf{b}'$ such that $\|\mathbf{b}'\|_p = \mathcal{O}(\text{OPT})$.

**Partitioning the matrix into groups.** We then round the entries of $\mathbf{b}'$ to obtain a vector $\mathbf{b}''$ with at most $\ell := \mathcal{O}\left(\frac{\log n}{\varepsilon}\right)$ unique values, truncating all entries that are less in magnitude than $\frac{1}{\text{poly}(n)}$ to zero instead. We now solve the $\ell_p$ regression problem $\min_{\mathbf{x} \in \mathbb{R}^d} \|\mathbf{A}\mathbf{x} - \mathbf{b}''\|_p$ by partitioning the rows of $\mathbf{A}$ into $\ell$ groups $G_1, \ldots, G_\ell$ based on their corresponding values of $\mathbf{b}''$. The main point is that all rows in each group $G_k$ have the same entry $t_k$ in $\mathbf{b}''$. Thus we can again observe that $|\langle \mathbf{a}_i, \mathbf{x} \rangle - t_k|^p$ can be written as a linear combination of $\mathcal{O}(p^2 d)$ variables and therefore use $\ell_q$ Lewis weight sampling (for the same $q$) to reduce each group $G_k$ down to $\mathcal{O}\left(\frac{p^2 d}{\varepsilon^2} \log d\right)$ rows. Since there are $\ell = \mathcal{O}\left(\frac{\log n}{\varepsilon}\right)$ groups, then there are roughly $\mathcal{O}(p^2 d)$ total rows that have been sampled across all the groups. It follows from the decomposition of the $\ell_p$ loss function across each group that the matrix $\mathbf{T}$ formed by these rows is a coreset for the $\ell_p$ regression problem. Therefore, by solving the $\ell_p$ regression problem on $\mathbf{T}$, which has significantly smaller dimension than the original input matrix $\mathbf{A}$, we obtain a vector $\widehat{\mathbf{x}}$ with the desired property that

$$\|\mathbf{A}\widehat{\mathbf{x}} - \mathbf{b}\|_p \leq (1 + \mathcal{O}(\varepsilon)) \min_{\mathbf{x} \in \mathbb{R}^d} \|\mathbf{A}\mathbf{x} - \mathbf{b}\|_p.$$

**Practical $\ell_p$ regression for arbitrary matrices.** We now describe a practical procedure for $\ell_p$ regression on general matrices that avoids the necessity for convex programming to approximate the $\ell_p$ Lewis weights for $p > 4$. We first pick an integer $r$ such that $\frac{p}{2^r} \in [2, 4)$. By the same structural argument as in Lemma 1.4, we can tensor product each row $\mathbf{a}_i$ of $\mathbf{A}$ with itself $2^r$ times, thus obtaining an extended matrix of size $n \times d'$, where $d' = d^{2^r}$, independent of the entries in $\mathbf{b}$. We then $\ell_{p/2^r}$ Lewis weight sample on the extended matrix, using the iterative method in Figure 1 since $\frac{p}{2^r} < 4$ rather than solving a convex program for $\ell_p$ Lewis weight sampling for $p > 4$. Since $\frac{p}{2^r} \in [2, 4)$, it follows from Theorem 2.3 that Lewis weight sampling requires roughly $(d')^{p/2}$ rows. Thus we obtain a matrix with $\mathcal{O}\left((d^{2^r})^{p/2^{r+1}}\right) = \mathcal{O}\left(d^{p/2}\right)$ rows, from which we can compute a residual vector $\mathbf{b}'$ such that $\|\mathbf{b}'\|_p = \mathcal{O}(\text{OPT})$, where $\text{OPT} := \min_{\mathbf{x} \in \mathbb{R}^d} \|\mathbf{A}\mathbf{x} - \mathbf{b}\|_p$.

We then round and truncate the entries of $\mathbf{b}'$ using the subroutine RoundTrunc to obtain a vector $\mathbf{b}''$, which allows us to partition the rows of $\mathbf{A}$ and the entries of $\mathbf{b}''$ into $\mathcal{O}\left(\frac{\log n}{\varepsilon}\right)$ groups. Due to the constant values of $\mathbf{b}''$ in each group, we can again $\ell_{p/2^r}$ Lewis weight sample on an extended matrix using an iterative method and finally solve the $\ell_p$ regression problem on the subsequent rows that have been sampled. We give our algorithm in full in Algorithm 3.

## 2  $\ell_p$ REGRESSION FOR VANDERMONDE MATRICES

### 2.1  PRELIMINARIES ON LEWIS WEIGHT SAMPLING

There are a number of known sampling distributions for dimensionality reduction for the $\ell_p$ regression problem, such as the $\ell_p$ leverage scores, e.g., (Cormode et al., 2018; Dasgupta et al., 2008) and the $\ell_p$ sensitivities, e.g., (Clarkson et al., 2019; Braverman et al., 2020; 2021; Musco et al., 2021); in this paper we focus on the $\ell_p$ Lewis weights, e.g., (Cohen & Peng, 2015; Durfee et al., 2018; Chen & Derezinski, 2021; Parulekar et al., 2021).

**Definition 2.1** ($\ell_p$ Lewis Weights, (Cohen & Peng, 2015))**.** *Given a matrix $\mathbf{A} \in \mathbb{R}^{n \times d}$ and $p \geq 1$, the $\ell_p$ Lewis weights $w_1(\mathbf{A}), \ldots, w_n(\mathbf{A})$ are the **unique** quantities that satisfy*

$$(w_i(\mathbf{A}))^{2/p} = \mathbf{a}_i^\top (\mathbf{A}^\top \mathbf{W}^{1-2/p} \mathbf{A})^{-1} \mathbf{a}_i$$

*for all $i \in [n]$, where $\mathbf{W} \in \mathbb{R}^{n \times n}$ is the diagonal matrix with $W_{i,i} = w_i(\mathbf{A})$ for all $i \in [n]$.*

**Definition 2.2** (Lewis Weight Sampling). *For an input matrix $\mathbf{A} \in \mathbb{R}^{n \times d}$ and $m$ samples, let the sampling matrix $\mathbf{S} \in \mathbb{R}^{m \times n}$ be generated by independently setting each row of $\mathbf{S}$ to be the $i$-th standard basis vector multiplied by $\left( \frac{d}{m \cdot w_i^p(\mathbf{A})} \right)^{1/p}$ with probability $\frac{w_i^p(\mathbf{A})}{d}$.*

**Theorem 2.3** ($\ell_p$ Subspace Embedding from Lewis Weight Sampling, (Cohen & Peng, 2015)). *For any $\mathbf{A} \in \mathbb{R}^{n \times d}$, let the matrix $\mathbf{S} \in \mathbb{R}^{m \times n}$ be generated from Lewis weight sampling with $m = \mathcal{O}\left( \frac{d^{\max(1, p/2)} \log(d/\delta) \log(1/\varepsilon)}{\varepsilon^{\gamma}} \right)$, where $\gamma = 2$ for $p \in [1, 2]$ and $\gamma = 5$ for $p > 2$. Then with probability at least $1 - \delta$, we have that simultaneously for all $\mathbf{x} \in \mathbb{R}^d$,*

$$(1 - \varepsilon)\|\mathbf{A}\mathbf{x}\|_p \leq \|\mathbf{S}\mathbf{A}\mathbf{x}\|_p \leq (1 + \varepsilon)\|\mathbf{A}\mathbf{x}\|_p.$$

Since the Lewis weights are implicitly defined, it may not be clear how to compute them exactly. In fact, it suffices to compute constant factor approximations to the Lewis weights. (Cohen & Peng, 2015) show that for $p \leq 4$, there exists a simple iterative approach to compute a constant factor approximation to the $\ell_p$ Lewis weights in input sparsity time, which we present in Figure 1.

---

$\mathbf{w} = \mathsf{LewisIterate}(\mathbf{A}, p, \beta, \mathbf{w})$

    (1) For $i = 1, \ldots, n$

        (a) Let $\tau_i$ be a constant factor approximation to $\mathbf{a}_i^\top (\mathbf{A}^\top \mathbf{W}^{1-2/p} \mathbf{A})^{-1} \mathbf{a}_i$

        (b) $w_i \leftarrow (\tau_i)^{p/2}$

        (c) Return $\mathbf{w}$

$\mathbf{w} = \mathsf{ApproxLewisWeights}(\mathbf{A}, p, \beta, T)$

    (1) Intialize $w_i = 1$ for all $i \in [n]$.

    (2) For $t = 1, \ldots, T$

        (a) $\mathbf{w} \leftarrow \mathsf{LewisIterate}(\mathbf{A}, p, \beta, \mathbf{w})$

    (3) Return $\mathbf{w}$

---

Fig. 1: Iterative algorithm for approximate $\ell_p$ Lewis weights with $p < 4$.

At a high level, the correctness of Figure 1 follows from Banach's fixed point theorem and the fact that the subroutine LewisIterate is a contraction mapping, because $|2/p - 1| < 1$ for $p < 4$ (Cohen & Peng, 2015). However, for $p \geq 4$, Figure 1 no longer works. Instead, prior to the recent work of (Fazel et al., 2021), approximating $\ell_p$ Lewis weights for $p > 4$ seems to require solving the convex program

$$\mathbf{Q} = \underset{\mathbf{M}}{\arg\max} \det(\mathbf{M}), \qquad \text{subject to } \sum_i (\mathbf{a}_i^\top \mathbf{M} \mathbf{a})^{p/2} \leq d, \qquad \mathbf{M} \succeq 0,$$

and setting $w_i = (\mathbf{a}_i^\top \mathbf{Q} \mathbf{a})^{p/2}$. Unfortunately, this is often infeasible in practice, and we could not obtain empirical results using it. Therefore, a nice advantage of our algorithms, both for structured and unstructured matrices, is that we only use $\ell_q$ Lewis weight sampling for $q \leq [1, 4)$, even if $p \geq 4$, whereas previous algorithms required using $\ell_p$ Lewis weight sampling for $p > 4$.

## 2.2 Algorithm and Analysis

We first describe the general framework of our algorithm for efficient $\ell_p$ regression, so that given $\mathbf{A} \in \mathbb{R}^{n \times d}$ and $\mathbf{b} \in \mathbb{R}^d$, the goal is to approximately compute $\mathrm{OPT} := \min_{\mathbf{x} \in \mathbb{R}^d} \|\mathbf{A}\mathbf{x} - \mathbf{b}\|_p$. Recall that in order to apply our structural results, we first require a measurement vector $\mathbf{b}''$ with a small number of distinct entries. We obtain $\mathbf{b}''$ by first finding a constant factor approximation, i.e., $\tilde{\mathbf{x}} \in \mathbb{R}^d$ such that $\|\mathbf{A}\tilde{\mathbf{x}} - \mathbf{b}\|_p \leq \mathcal{O}(\mathrm{OPT})$. We can $\ell_p$ Lewis weight sample the rows of $\mathbf{A}$ to do this, but the time to solve the subsequent polynomial regression problem would have exponential dependence on $p$, due to Theorem 2.3. Instead, we use our structural properties to implicitly create a matrix $\mathbf{M}$ from $\mathbf{A}$ with fewer than $d^p$ columns and then $\ell_{p/2^r}$ Lewis weight sample the rows of

$\mathbf{A}$, where $r$ is the integer that satisfies $2^r \leq p < 2^{r+1}$, and then we solve the $\ell_p$ regression problem on the sampled rows of $\mathbf{A}$ and entries of $\mathbf{b}$ to obtain $\tilde{\mathbf{x}}$. We set $\mathbf{b}'$ to be the residual vector $\mathbf{b} - \mathbf{A}\tilde{\mathbf{x}}$ so that $\|\mathbf{b}'\|_p = \mathcal{O}(\text{OPT})$. We then use the procedure RoundTrunc, i.e., as in Algorithm 1, which sets the entries of $\mathbf{b}'$ that are the smallest in magnitude to zero, and rounds the remaining entries of $\mathbf{b}'$ to the nearest power of $(1 + \varepsilon)$.

---

**Algorithm 1** RoundTrunc: Round and truncate coordinates of input vector $\mathbf{b}$

---

**Input:** Vector $\mathbf{b} \in \mathbb{R}^n$, accuracy parameter $\varepsilon > 0$
**Output:** Vector $\mathbf{x} \in \mathbb{R}^n$ that is a rounded and truncated version of $\mathbf{b}$
1: $M \leftarrow \max_{i \in n} |b_i|$
2: **for** $i = 1$ to $i = n$ **do**
3:      $p_i \leftarrow \lfloor \log_{1+\varepsilon} |b_i| \rfloor$
4:      $x_i \leftarrow (1 + \varepsilon)^{p_i} \cdot \text{sign}(b_i)$
5:      **if** $|x_i| \leq \frac{M}{n^5}$ **then**
6:          $x_i \leftarrow 0$
7: **return** $\mathbf{x}$

---

Since $\|\mathbf{b}'\|_p = \mathcal{O}(\text{OPT})$, it then follows by triangle inequality that it suffices to approximately solve the $\ell_p$ regression problem on the vector $\mathbf{b}''$ instead. We partition the rows of $\mathbf{A}$ into groups $G_1, \ldots, G_\ell$ based on the values of $\mathbf{b}''$ and perform $\ell_{p/2^r}$ Lewis weight sampling again on an implicit matrix that we create from the rows of each group. Here we leverage our theory about the rank of the $\ell_p$ regression problem for structured inputs. It then suffices to solve the $\ell_p$ regression problem on the matrix formed by the sampled rows across all the groups.

---

**Algorithm 2** Faster regression for Vandermonde matrices

---

**Input:** Vandermonde matrix $\mathbf{A} \in \mathbb{R}^{n \times d}$, measurement vector $\mathbf{b}$, accuracy parameter $\varepsilon > 0$
**Output:** $\widehat{\mathbf{x}} \in \mathbb{R}$ with $\|\mathbf{A}\widehat{\mathbf{x}} - \mathbf{b}\|_p \leq (1 + \mathcal{O}(\varepsilon)) \min_{\mathbf{x} \in \mathbb{R}^d} \|\mathbf{A}\mathbf{x} - \mathbf{b}\|_p$
1: $r \leftarrow \lfloor \log p \rfloor$                                                    $\triangleright 2^r \leq p < 2^{r+1}$.
2: Extend $\mathbf{A}$ to a Vandermonde matrix $\mathbf{M}$ with dimensions $n \times d'$, where $d' = (2^r(d-1) + 1)$.
3: Use $\ell_{p/2^r}$-Lewis weight sampling on $\mathbf{M}$ to find a set $S$ of $\mathcal{O}(d' \log d')$ indices in $[n] = \{1, 2, \ldots, n\}$, and corresponding rescaling factors.
4: Let $\mathbf{A}'$ be the corresponding submatrix of $\mathbf{A}$ with indices in $S$, and scaled accordingly.
5: Compute $\tilde{\mathbf{x}} \leq 5 \min_{\mathbf{x} \in \mathbb{R}^d} \|\mathbf{A}'\mathbf{x} - \mathbf{b}\|_p$.
6: $\mathbf{b}' \leftarrow \mathbf{b} - \mathbf{A}\tilde{\mathbf{x}}$, $\mathbf{b}'' \leftarrow \text{RoundTrunc}(\mathbf{b}')$, $\ell \leftarrow \mathcal{O}\left(\frac{\log n}{\varepsilon}\right)$
7: Partition the rows of $\mathbf{A}$ into groups $G_1, \ldots, G_\ell$, each containing all rows with the same value of $\mathbf{b}''$
8: Let $\mathbf{G}_k$ be the submatrix of $G_k$ extended to $d'' = 2^{2r}(d-1) + 1$ columns.
9: Let $t_k$ be the coordinate of $\mathbf{b}''$ corresponding to $G_k$ for each $k \in [\ell]$.
10: Use $\ell_{p/2^r}$-Lewis weight sampling on $[\mathbf{G}_k; t_k]$ to find a set $S'_k$ of $\mathcal{O}\left(\frac{d''}{\varepsilon^2} \log d''\right)$ indices in $[n]$ and rescaling factors.
11: Let $\mathbf{T}_k$ be the corresponding sampling and rescaling matrix for $S'_k$.
12: $\mathbf{T} \leftarrow [\mathbf{T}_1; \ldots; \mathbf{T}_k]^\top$
13: Compute $\widehat{\mathbf{x}} \leq (1 + \varepsilon) \min_{\mathbf{x} \in \mathbb{R}^d} \|\mathbf{T}\mathbf{A}\mathbf{x} - \mathbf{T}\mathbf{b}\|_p$.
14: **return** $\widehat{\mathbf{x}}$

---

We show in the supplementary material that Algorithm 2 satisfies the guarantees of Theorem 1.1. Moreover, because Theorem 1.1 has polynomial dependence on $p$ rather than exponential dependence, we obtain the first known sublinear size coreset for the important problem of $\ell_\infty$ regression. We use the following structural property.

**Lemma 2.4.** *Let $\mathbf{x} \in \mathbb{R}^n$ and $p = \Omega\left(\frac{\log n}{\varepsilon}\right)$. Then $\|\mathbf{x}\|_\infty \leq \|\mathbf{x}\|_p \leq (1 + \varepsilon)\|\mathbf{x}\|_\infty$.*

Lemma 2.4 implies that to solve $\ell_\infty$ regression, we can instead solve $\ell_p$ regression for $p = \Omega\left(\frac{\log n}{\varepsilon}\right)$. Then Theorem 1.2 follows from the fact that even for $p = \Omega\left(\frac{\log n}{\varepsilon}\right)$, the matrix $\mathbf{T}$

in Algorithm 2 satisfies

$$(1 - \varepsilon)\|\mathbf{T}\mathbf{A}\mathbf{x} - \mathbf{T}\mathbf{b}''\|_p \leq \|\mathbf{A}\mathbf{x} - \mathbf{b}''\|_p \leq (1 + \varepsilon)\|\mathbf{T}\mathbf{A}\mathbf{x} - \mathbf{T}\mathbf{b}''\|_p$$

for all $\mathbf{x} \in \mathbb{R}^d$. Hence, we can solve the $\ell_p$ regression problem on the smaller matrix $\mathbf{T}$ to solve the $\ell_\infty$ regression on $\mathbf{A}$.

The results of Theorem 1.1 can be further extended to matrices with block Vandermonde structure.

**Corollary 2.5.** *Given $\varepsilon \in (0, 1)$ and $p \geq 1$, $\mathbf{A} \in \mathbb{R}^{n \times dq}$, and $\mathbf{b} \in \mathbb{R}^d$, suppose $\mathbf{A} = [\mathbf{A}_1 | \ldots | \mathbf{A}_q]$ for Vandermonde matrices $\mathbf{A}_1, \ldots, \mathbf{A}_q \in \mathbb{R}^{n \times d}$. Then there exists an algorithm that, with high probability, returns a vector $\widehat{\mathbf{x}} \in \mathbb{R}^d$ such that*

$$\|\mathbf{A}\widehat{\mathbf{x}} - \mathbf{b}\|_p \leq (1 + \varepsilon) \min_{\mathbf{x} \in \mathbb{R}^d} \|\mathbf{A}\mathbf{x} - \mathbf{b}\|_p,$$

*using $\mathcal{O}\left(T(\mathbf{A})(dp)^{q-1} \log n + \text{poly}((dp)^q \log n, 1/\varepsilon)\right)$ time, where $T(\mathbf{A})$ is the runtime of multiplying the matrix $\mathbf{A}$ by an arbitrary vector. For $q = 2$, this can be further optimized to $\mathcal{O}\left(nd^{\omega_2/2-1} + \text{poly}((dp)^2, \log n, 1/\varepsilon)\right)$ time, where $\mathcal{O}(n^{\omega_2})$ is the time to multiply an $n \times n$ matrix with an $n \times n^2$ matrix, so that $\omega_2 \in [3, 4]$.*

We obtain similar algorithms for noisy low-rank matrices and noisy Vandermonde matrices, i.e., Theorem 1.5 and Theorem 1.6. We defer presentation and justification for these results to the supplementary material.

---

**Algorithm 3** Faster $\ell_p$ regression for general matrices

---

**Input:** Matrix $\mathbf{A} \in \mathbb{R}^{n \times d}$, measurement vector $\mathbf{b}$, accuracy parameter $\varepsilon > 0$
**Output:** $\widehat{\mathbf{x}} \in \mathbb{R}$ with $\|\mathbf{A}\widehat{\mathbf{x}} - \mathbf{b}\|_p \leq (1 + \varepsilon) \min_{\mathbf{x} \in \mathbb{R}^d} \|\mathbf{A}\mathbf{x} - \mathbf{b}\|_p$, parameter $p \geq 4$
 1: $r \leftarrow \lfloor \log p \rfloor - 1$ $\qquad\qquad\qquad\qquad\qquad\qquad\qquad\qquad\qquad \triangleright 2^{r+1} \leq p < 2^{r+2}$.
 2: Extend $\mathbf{A}$ to a matrix $\mathbf{M}$ with dimension $n \times \mathcal{O}\left(d^{2^r}\right)$ so that each row in $M_i$ is the $2^r$-fold tensor product of $\mathbf{a}_i$ reshaped into a $d^{2^r}$ length vector.
 3: Use $\ell_{p/2^r}$-Lewis weight sampling on $\mathbf{M}$ to find a set $S$ of $\mathcal{O}\left(d^{p/2} \log d\right)$ indices in $[n]$ and rescaling factors.
 4: Let $\mathbf{A}'$ be the corresponding submatrix of $\mathbf{A}$ with indices in $S$ and scaled accordingly.
 5: Compute $\tilde{\mathbf{x}} \leq 5 \min_{\mathbf{x} \in \mathbb{R}^d} \|\mathbf{A}'\mathbf{x} - \mathbf{b}\|_p$.
 6: $\mathbf{b}' \leftarrow \mathbf{b} - \mathbf{A}\tilde{\mathbf{x}}$, $\mathbf{b}'' \leftarrow \mathsf{RoundTrunc}(\mathbf{b}')$, $\ell \leftarrow \mathcal{O}\left(\frac{\log n}{\varepsilon}\right)$
 7: Partition the rows of $\mathbf{A}$ into groups $G_1, \ldots, G_\ell$, each containing all rows with the same value of $\mathbf{b}''$
 8: Let $\mathbf{G}_k$ be the corresponding submatrix and $t_k$ be the coordinate of $\mathbf{b}''$ corresponding to $G_k$ for each $k \in [\ell]$.
 9: Use $\ell_{p/2^r}$-Lewis weight sampling on $[\mathbf{G}_k; t_k]$ to find a set $S'_k$ of $\mathcal{O}\left(\frac{1}{\varepsilon^5} d^{p/2} \log d\right)$ indices in $[n]$ and rescaling factors.
10: Let $\mathbf{T}_k$ be the corresponding rows with indices in $S'_k$ and scaled accordingly.
11: $\mathbf{T} \leftarrow [\mathbf{T}_1; \ldots; \mathbf{T}_k]^\top$
12: Compute $\widehat{\mathbf{x}} \leq (1 + \varepsilon) \min_{\mathbf{x} \in \mathbb{R}^d} \|\mathbf{T}\mathbf{x} - \mathbf{T}\mathbf{b}\|_p$.
13: **return** $\widehat{\mathbf{x}}$

---

Finally, we describe in Algorithm 3 a practical approach for $\ell_p$ regression for arbitrary matrices *without requiring any structural assumptions.*

**Theorem 2.6.** *Algorithm 3 outputs a vector $\widehat{\mathbf{x}}$ such that $\|\mathbf{A}\widehat{\mathbf{x}} - \mathbf{b}\|_p \leq (1 + \varepsilon) \min \|\mathbf{A}\mathbf{x} - \mathbf{b}\|_p$. The runtime of Algorithm 3 is $T(n, d^{p/2}) + \text{poly}\left(d^{p/2}, \log n, \frac{1}{\varepsilon}\right)$, where $T(n, d^{p/2})$ is the time to multiply a matrix of size $n \times d^{p/2}$ by a vector of length $d^{p/2}$.*

Theorem 2.6 achieves the same optimal sample complexity as previous $\ell_p$ Lewis weight algorithms of roughly $d^{p/2}$. However, the main advantage of Algorithm 3 is that it performs $\ell_q$ Lewis weight sampling for some $q \in [2, 4)$, which is quite efficient because we can use an iterative method rather than solving a convex program.

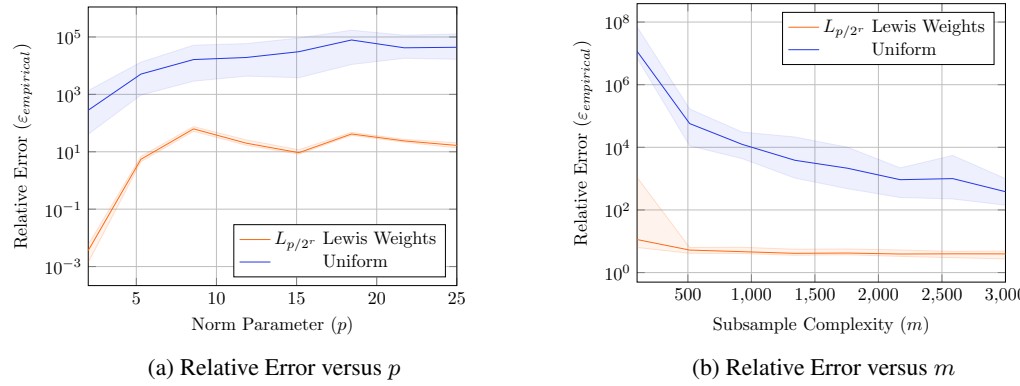

(a) Relative Error versus $p$  (b) Relative Error versus $m$

Fig. 2: Comparison of relative error $(\varepsilon_{empirical})$ versus $\ell_p$ parameter $p$ and sample complexity $m$ on Vandermonde data. We ran both experiments 30 times and plot the median, $25^{th}$ quartile, and $75^{th}$ quartile for each value of $p$ and $m$.

## 3 EMPIRICAL VERIFICATION

We provide empirical evidence to validate our core statistical claim about Vandermonde regression: that the relative error $\varepsilon$ achieved by $\ell_{p/2^r}$ Lewis Weight subsampling is polynomial in $p$, and not exponential in $p$. More precisely, we compute the gap between the error achieved from exact $\ell_p$ Vandermonde regression and the error achieved from the subsampled regression:

$$\varepsilon_{empirical} = \frac{\|\mathbf{V}\widehat{\mathbf{x}} - \mathbf{b}\|_p - \|\mathbf{V}\mathbf{x}^* - \mathbf{b}\|_p}{\|\mathbf{V}\mathbf{x}^* - \mathbf{b}\|_p}$$

Our theory tells us that $(\varepsilon_{empirical})^3 \leq \tilde{\mathcal{O}}\left(\frac{dp^2}{m}\right)$, where $m$ is the total number of subsampled rows. The prior work on unstructured matrices instead suggests $(\varepsilon_{empirical})^c \leq \tilde{\mathcal{O}}\left(\frac{d^{\mathcal{O}(p)}}{m}\right)$ (Shi & Woodruff, 2019). So, to visually distinguish these two settings, we look at the logarithm of both sides:

$$\log(\varepsilon_{empirical}) \leq \mathcal{O}\left(\ln(p) + \ln(\tfrac{d}{m})\right) \qquad \text{OR} \qquad \log(\varepsilon_{empirical}) \leq \mathcal{O}\left(p\ln(d) + \ln(\tfrac{1}{m})\right)$$

In particular, our work suggests a *logarithmic* dependence on $p$, while the prior work suggests a *linear* dependence on $p$.

To validate our theory, we plot $\log(\varepsilon_{empirical})$ versus $p$ and $m$ on synthetic data. Specifically, we generate $n = 25,000$ i.i.d. $N(0,1)$ times samples to form a Vandermonde matrix $\mathbf{V}$ with $d = 20$ columns, then compute the polynomial $q(t) = t^{10}$ at each time sample, add $N(0, 10^{10})$ additive noise, and save the corresponding values in $\mathbf{b}$. We then compute $\varepsilon_{empirical}$ for this $\ell_p$ regression problem. Notably, in order to compute $\widehat{\mathbf{x}}$, we omit the rounding procedure in our code, since the rounding is designed for worst-case inputs. Instead, we simply compute $\widehat{\mathbf{x}}$ by solving $\min_{\mathbf{x}} \|\widehat{\mathbf{V}}\mathbf{x} - \widehat{\mathbf{b}}\|_p$ where $\widehat{\mathbf{V}}$ and $\widehat{\mathbf{b}}$ are computed by sampling and rescaling $\mathbf{V}$ and $\mathbf{b}$ with the $\ell_{p/2^r}$ Lewis Weights.

Figure 2 shows the result of these experiments, which were run in Julia 1.6.1, on Windows 10 with an Intel i7-7700K CPU and 16Gb RAM. In Figure 2a, we fix $m = 1000$ and vary $p \in [2, 25]$. The trendline of Lewis Weight sampling clearly better fits a logarithmic model, as opposed to a linear model. This reinforces our analysis by showing that the dependence on $p$ is notably sub-exponential, beating the known bounds for $\ell_p$ subsampling on unstructured matrices.

As a benchmark, we compare our Lewis Weight sampling method to uniform sampling. The noise in $\mathbf{b}$ is large enough that most rows of $\mathbf{A}$ have little information about the underlying polynomial $q(t)$. Lewis weight sampling takes avoids these rows, while uniform sampling does not, explaining why uniform sampling is much weaker in Figure 2a. Further, in Figure 2b, we fix $p = 6$ and vary $m \in [100, 3000]$, showing that Lewis Weight sampling outperforms uniform sampling across both $m$ and $p$. In Appendix C, we demonstrate similar results for unstructured matrix regression, validating the analysis of Theorem 2.6.

## ACKNOWLEDGEMENTS

Cameron Musco was supported by NSF grants 2046235 and 1763618, and an Adobe Research grant. David P. Woodruff and Samson Zhou were supported by National Institute of Health grant 5401 HG 10798-2 and a Simons Investigator Award. Christopher Musco and Raphael Meyer were supported by NSF grant 2045590 and DOE Award DE-SC0022266.

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

## BROADER IMPACT

This work is focused on improving the runtime of algorithms for $\ell_p$ regression on structured inputs, and can allow users to more efficiently solve regression problems on large datasets. The problem we study is abstracted away from specific ethical and unethical applications of $\ell_p$ regression. So, *with the assumption that the research and data science communities will **focus on** ethical applications while **avoiding** unethical ones*, we believe that the benefits of our algorithmic results outweigh the negative risks.

## A   MISSING PROOFS FROM SECTION 2

In this section, we give the missing proofs from Section 2.

### A.1   $\ell_p$ REGESSION ON VANDERMONDE MATRICES

We first prove a simple statement that shows a "good" solution to the $\ell_p$ regression problem on a subspace embedding $\mathbf{SA}$ is also a "good" solution to the original input matrix $\mathbf{A}$.

**Lemma A.1.** *Let $\mathbf{S}$ be a sampling and rescaling matrix such that*

$$\frac{11}{12}\|\mathbf{SAx}\|_p \leq \|\mathbf{Ax}\|_p \leq \frac{13}{12}\|\mathbf{SAx}\|_p, \qquad \mathbb{E}\left[\|\mathbf{SAx}\|_p^p\right] = \|\mathbf{Ax}\|_p^p$$

*for all $\mathbf{x} \in \mathbb{R}^d$. Let $OPT = \min_{\mathbf{x} \in \mathbb{R}^d} \|\mathbf{Ax} - \mathbf{b}\|_p$ and let $\mathbf{x} \in \mathbb{R}^d$ be any vector for which $\|\mathbf{SAx} - \mathbf{Sb}\|_p \leq 5OPT$. Then with probability at least $0.79$,*

$$\|\mathbf{Ax} - \mathbf{b}\|_p \leq 12OPT.$$

*Proof.* Let $\mathbf{x}^*$ be a minimizer of $\|\mathbf{Ax} - \mathbf{b}\|_p$ so that $OPT = \|\mathbf{Ax}^* - \mathbf{b}\|_p$. We will prove the claim by contrapositive, so we first suppose that $\|\mathbf{Ax} - \mathbf{b}\|_p \leq 12OPT$. Then by the triangle inequality,

$$\|\mathbf{SAx} - \mathbf{Sb}\|_p \geq \|\mathbf{SA}(\mathbf{x} - \mathbf{x}^*)\|_p - \|\mathbf{SA}^* - \mathbf{Sb}\|_p.$$

Since $\frac{11}{12}\|\mathbf{SAx}\|_p \leq \|\mathbf{Ax}\|_p \leq \frac{13}{12}\|\mathbf{SAx}\|_p$ for all $\mathbf{x} \in \mathbb{R}^d$, then

$$\|\mathbf{SAx} - \mathbf{Sb}\|_p \geq \frac{11}{12}\|\mathbf{A}(\mathbf{x} - \mathbf{x}^*)\|_p - \|\mathbf{SA}^* - \mathbf{Sb}\|_p.$$

By the triangle inequality,

$$\|\mathbf{S}\mathbf{A}\mathbf{x} - \mathbf{S}\mathbf{b}\|_p \geq \frac{11}{12} \left( \|\mathbf{A}\mathbf{x} - \mathbf{b}\|_p - \|\mathbf{A}\mathbf{x}^* - \mathbf{b}\|_p \right) - \|\mathbf{S}\mathbf{A}^* - \mathbf{S}\mathbf{b}\|_p.$$

Note that by Jensen's inequality, $\mathbb{E}[\|\mathbf{S}\mathbf{A}\mathbf{x}^* - \mathbf{S}\mathbf{b}\|_p] \leq \text{OPT}$, so that by Markov's inequality,

$$\mathbf{Pr}\left[\|\mathbf{S}\mathbf{A}\mathbf{x}^* - \mathbf{S}\mathbf{b}\|_p \geq 5\text{OPT}\right] \leq \frac{1}{5}.$$

Thus with probability at least 0.8,

$$\|\mathbf{S}\mathbf{A}\mathbf{x} - \mathbf{S}\mathbf{b}\|_p \geq \frac{11}{12} \left( \|\mathbf{A}\mathbf{x} - \mathbf{b}\|_p - \|\mathbf{A}\mathbf{x}^* - \mathbf{b}\|_p \right) - 5\text{OPT}.$$

Thus if $\|\mathbf{A}\mathbf{x} - \mathbf{b}\|_p \leq 12\text{OPT}$, then

$$\|\mathbf{S}\mathbf{A}\mathbf{x} - \mathbf{S}\mathbf{b}\|_p \geq \frac{11}{12} \left( 12\text{OPT} - \text{OPT} \right) - 5\text{OPT} > 5\text{OPT},$$

as desired. $\qquad\square$

Before justifying the correctness of Algorithm 2, we first recall the following algorithm for efficient $\ell_p$ subspace embeddings.

**Theorem A.2.** *(Shi & Woodruff, 2019) Given $\varepsilon \in (0,1)$ and $p \in [1,2]$, a Vandermonde matrix $\mathbf{A} \in \mathbb{R}^{n \times d}$, and $\mathbf{b} \in \mathbb{R}^d$, let $T(\mathbf{A})$ be the time it takes to perform matrix-vector multiplication, i.e., compute $\mathbf{A}\mathbf{v}$ for an arbitrary $\mathbf{v} \in \mathbb{R}^d$. There exists an algorithm that uses $\mathcal{O}\left(T(\mathbf{A})\log n + d^q \cdot \text{poly}(1/\varepsilon)\right)$ time, where $q = \omega$ for $p \in [1,4)$ and $q = p/2 + C$ for some fixed constant $C > 0$ for $p > 4$, and with high probability, returns $\tilde{\mathbf{x}} \in \mathbb{R}^d$ such that*

$$\|\mathbf{A}\tilde{\mathbf{x}} - \mathbf{b}\|_p \leq (1 + \varepsilon) \min_{\mathbf{x} \in \mathbb{R}^d} \|\mathbf{A}\mathbf{x} - \mathbf{b}\|_p.$$

We now justify the correctness of Algorithm 2.

**Lemma A.3.** *Given $\varepsilon \in (0,1)$ and $p \geq 1$, a Vandermonde matrix $\mathbf{A} \in \mathbb{R}^{n \times d}$, and $\mathbf{b} \in \mathbb{R}^d$, then with high probability, Algorithm 2 returns a vector $\widehat{\mathbf{x}} \in \mathbb{R}^d$ such that*

$$\|\mathbf{A}\widehat{\mathbf{x}} - \mathbf{b}\|_p \leq (1 + \mathcal{O}(\varepsilon)) \min_{\mathbf{x} \in \mathbb{R}^d} \|\mathbf{A}\mathbf{x} - \mathbf{b}\|_p.$$

*Proof.* Consider Algorithm 2 and let $r$ be an integer so that $2^r \leq p < 2^{r+1}$. Note that

$$\sum_{i \in [n]} |\langle \mathbf{a}_i, \mathbf{x} \rangle|^p = \sum_{i \in [n]} |(\langle \mathbf{a}_i, \mathbf{x} \rangle)^{2^r}|^{p/2^r}.$$

Since $\mathbf{A}$ is Vandermonde, then

$$(\langle \mathbf{a}_i, \mathbf{x} \rangle)^{2^r} = \left( \sum_{j=1}^d a_{i,j} x_j \right)^{2^r} = \left( \sum_{j=1}^d a_{i,1}^j x_j \right)^{2^r} = \sum_{j=1}^{2^r(d-1)+1} a_{i,2}^{j-1} y_j,$$

where each $y_j$ is a fixed function of the coordinates of $\mathbf{x}$. Notably, the fixed function *is the same across all $i \in [n]$*. Hence, the $\ell_{2^r}$ subspace embedding problem on an input Vandermonde matrix $\mathbf{A} \in \mathbb{R}^{n \times d}$ can be reshaped as a constrained $\ell_1$ subspace embedding problem on an input Vandermonde matrix of size $n \times (2^r(d-1)+1)$. Thus, for $\ell_p$ regression with $p \in [2^r, 2^{r+1})$, we have

$$\sum_{i \in [n]} |\langle \mathbf{a}_i, \mathbf{x} \rangle|^p = \sum_{i \in [n]} \left| \sum_{j=1}^{2^r(d-1)+1} a_{i,2}^{j-1} y_j \right|^{p/2^r},$$

which is a constrained $\ell_{p/2^r}$ regression problem on an input Vandermonde matrix $\mathbf{M}$ of size $n \times (2^r(d-1)+1)$.

By Theorem A.2, we can use $\ell_{p/2^r}$ Lewis weight sampling to find a matrix $\mathbf{M}'$ such that

$$\frac{1}{5}\|\mathbf{M}\mathbf{y}\|_{p/2^r}^{p/2^r} \leq \|\mathbf{M}'\mathbf{y}\|_{p/2^r}^{p/2^r} \leq 5\|\mathbf{M}\mathbf{y}\|_{p/2^r}^{p/2^r}$$

for all $\mathbf{y} \in \mathbb{R}^{2^r(d-1)+1}$ with high probability. Note that by the above argument, if we take the matrix $\mathbf{A}'$ corresponding to the scaled rows of $\mathbf{A}$ that are sampled by $\mathbf{M}'$, then we also have

$$\frac{11}{12}\|\mathbf{Ax}\|_p^p \le \|\mathbf{A}'\mathbf{x}\|_p^p \le \frac{13}{12}\|\mathbf{Ax}\|_p^p$$

and thus

$$\frac{11}{12}\|\mathbf{Ax}\|_p \le \|\mathbf{A}'\mathbf{x}\|_p \le \frac{13}{12}\|\mathbf{Ax}\|_p$$

for all $\mathbf{x} \in \mathbb{R}^{d+1}$ with high probability. Thus by Lemma A.1, we can find a vector $\tilde{\mathbf{x}}$ such that

$$\|\mathbf{A}\tilde{\mathbf{x}} - \mathbf{b}\|_p \le 12 \min_{\mathbf{x} \in \mathbb{R}^d} \|\mathbf{Ax} - \mathbf{b}\|_p.$$

We use the subroutine RoundTrunc to create the vector $\mathbf{b}''$ which is the vector with all entries of $\mathbf{b}'$ rounded to the nearest power of $(1 + \varepsilon)$, starting at the maximum entry of $\mathbf{b}'$ in absolute value, and stopping after we are $\frac{1}{\text{poly}(n)}$ times that, and replacing all remaining entries with $0$. By the triangle inequality, we have

$$\|\mathbf{Ax} - \mathbf{b}''\|_p \le \|\mathbf{Ax} - \mathbf{b}'\|_p + \|\mathbf{b}' - \mathbf{b}''\|_p \le \|\mathbf{Ax} - \mathbf{b}'\|_p + 12\varepsilon\text{OPT} \le (1 + 12\varepsilon)\|\mathbf{Ax} - \mathbf{b}''\|_p,$$

for any $\mathbf{x} \in \mathbb{R}^{d+1}$.

Note that $\mathbf{b}''$ has discretized the values of $\mathbf{b}'$ into $\ell = \mathcal{O}\left(\frac{\log n}{\varepsilon}\right)$ possible values. We partition the rows of $\mathbf{A}$ into $\ell$ groups $G_1, \ldots, G_\ell$, based on the corresponding values of $\mathbf{b}''$. Suppose that for a group $G_k$, the corresponding values of $\mathbf{b}''$ are all $t_k$. Then we have

$$\|\mathbf{Ax} - \mathbf{b}''\|_p^p = \sum_k \sum_{i \in G_k} |\langle \mathbf{a}_i, \mathbf{x}\rangle - t_k|^p.$$

Since $\mathbf{A}$ is Vandermonde, for each $i \in G_k$,

$$(\langle \mathbf{a}_i, \mathbf{x}\rangle - t_k)^{2^r} = \left(-t_k + \sum_{j=1}^d a_{i,j} x_j\right)^{2^r} = \left(-t_k + \sum_{j=1}^d a_{i,2}^{j-1} x_j\right)^{2^r} = \sum_{j=1}^{2^{2r}(d-1)+1} a_{i,2}^{j-1} t_{k,j} y_j,$$

for some fixed values $t_{k,1}, t_{k,2}, \ldots$ that can be computed from $t_k$, where again each $y_j$ can be a different function of the coordinates of $\mathbf{x}$. In particular, there can be $2^r$ choices for the exponent of $t_k$. For a fixed choice of the exponent of $t_k$, the exponent of $a_{i,2}$ can range from $0$ to $2^r(d-1)$.

Hence, the $\ell_{2^r}$ regression problem on a submatrix of an input Vandermonde matrix $\mathbf{A} \in \mathbb{R}^{n \times d}$ *with the same fixed measurement values*, i.e., the corresponding coordinates of $\mathbf{b}''$ are all the same, can be reshaped to a constrained $\ell_1$ regression problem on an input Vandermonde matrix with $2^{2r}(d-1)+1$ columns times a $(2^{2r}(d-1)+1) \times (2^{2r}(d-1)+1)$ diagonal matrix.

Hence, by invoking Theorem A.2 to sample rows of $\mathbf{M}$ corresponding to their $\ell_{p/2^r}$ Lewis weights in the submatrix induced by the rows of $G_k$, we obtain a sampling matrix $\mathbf{T}_k$ such that with high probability,

$$(1-\varepsilon)\|\mathbf{T}_k\mathbf{My} - \mathbf{T}_k\mathbf{v}_k\|_{p/2^r}^{p/2^r} \le \sum_{i \in G_k} \left(\sum_{j=1}^{2^r(d-1)+1} a_{i,2}^{j-1} t_{k,j} y_j\right)^{p/2^r} \le (1+\varepsilon)\|\mathbf{T}_k\mathbf{My} - \mathbf{T}_k\mathbf{v}_k\|_{p/2^r}^{p/2^r},$$

where $\mathbf{v}_k$ corresponds to the vector $\mathbf{b}''$ that is set to zero outside of coordinates whose values are $t_k$. Note that by the above argument, then we also have

$$(1 - \varepsilon)\|\mathbf{T}_k\mathbf{Ax} - \mathbf{T}_k\mathbf{v}_k\|_p^p \le \sum_{i \in G_k} |\langle \mathbf{a}_i, \mathbf{x}\rangle - t_k|^p \le (1 + \varepsilon)\|\mathbf{T}_k\mathbf{Ax} - \mathbf{T}_k\mathbf{v}_k\|_p^p,$$

with high probability. Summing over all $k$, we have

$$(1 - \varepsilon)\sum_k \|\mathbf{T}_k\mathbf{Ax} - \mathbf{T}_k\mathbf{v}_k\|_p^p \le \sum_k \sum_{i \in G_k} |\langle \mathbf{a}_i, \mathbf{x}\rangle - t_k|^p \le (1 + \varepsilon)\sum_k \|\mathbf{T}_k\mathbf{Ax} - \mathbf{T}_k\mathbf{v}_k\|_p^p,$$

with high probability.

Let $\mathbf{T}$ be the sampling matrix so that $\mathbf{T} = \mathbf{T}_1 \circ \ldots \circ \mathbf{T}_\ell$, so that $\sum_k \|\mathbf{T}_k \mathbf{A}\mathbf{x} - \mathbf{T}_k \mathbf{v}_k\|_p^p = \|\mathbf{T}\mathbf{A}\mathbf{x} - \mathbf{T}\mathbf{b}''\|_p^p$. Observe that $\|\mathbf{A}\mathbf{x} - \mathbf{b}''\|_p^p = \sum_k \sum_{i \in G_k} |\langle \mathbf{a}_i, \mathbf{x} \rangle - t_k|^p$. Hence,

$$(1 - \varepsilon)\|\mathbf{T}\mathbf{A}\mathbf{x} - \mathbf{T}\mathbf{b}''\|_p^p \leq \|\mathbf{A}\mathbf{x} - \mathbf{b}''\|_p^p \leq (1 + \varepsilon)\|\mathbf{T}\mathbf{A}\mathbf{x} - \mathbf{T}\mathbf{b}''\|_p^p$$

and thus

$$(1 - \varepsilon)\|\mathbf{T}\mathbf{A}\mathbf{x} - \mathbf{T}\mathbf{b}''\|_p \leq \|\mathbf{A}\mathbf{x} - \mathbf{b}''\|_p \leq (1 + \varepsilon)\|\mathbf{T}\mathbf{A}\mathbf{x} - \mathbf{T}\mathbf{b}''\|_p.$$

Thus we can compute a vector $\widehat{\mathbf{x}} \in \mathbb{R}^d$ such that

$$\|\mathbf{A}\widehat{\mathbf{x}} - \mathbf{b}\|_p \leq (1 + \mathcal{O}(\varepsilon)) \min_{\mathbf{x} \in \mathbb{R}^d} \|\mathbf{A}\mathbf{x} - \mathbf{b}\|_p.$$

$\square$

Before analyzing the time complexity of Algorithm 2, we first recall the following algorithms for Vandermonde matrix-vector multiplication and approximate $\ell_p$ regression.

**Theorem A.4** (Vandermonde Matrix-Vector Multiplication Runtime, e.g., Table 1 in (Gohberg & Olshevsky, 1994)). *The runtime of computing $\mathbf{A}\mathbf{x}$ for a Vandermonde matrix $\mathbf{A} \in \mathbb{R}^{n \times d}$ and a vector $\mathbf{x} \in \mathbb{R}^d$ for $d \leq n$ is $\mathcal{O}(n \log^2 n)$.*

**Theorem A.5** (Approximate $\ell_p$ Regression Runtime). *(Adil et al., 2019b) Given $\mathbf{A} \in \mathbb{R}^{n \times d}$ and $p \geq [2, \infty)$, there exists an algorithm that makes $\mathcal{O}\left(\sqrt{n} \log \frac{n}{\varepsilon}\right)$ calls to a linear system solver and computes a vector $\tilde{\mathbf{x}}$ such that*

$$\|\mathbf{A}\tilde{\mathbf{x}} - \mathbf{b}\|_p \leq (1 + \varepsilon) \min_{\mathbf{x} \in \mathbb{R}^d} \|\mathbf{A}\mathbf{x} - \mathbf{b}\|_p.$$

We now analyze the runtime of Algorithm 2.

**Lemma A.6.** *Algorithm 2 runs in $\mathcal{O}(n \log^3 n) + d^{0.5+\omega} \operatorname{poly}\left(\frac{1}{\varepsilon}, p, \log n\right)$ time.*

*Proof.* Observe that Algorithm 2 has three main bottlenecks for runtime. Since we only need to Lewis weight sample from the extended matrices, we do not need to explicitly form them, which would otherwise require $\Omega(ndp^2)$ time just to list to entries. Hence the first bottleneck is performing the Lewis weight sampling procedure on the extended matrices. The second bottleneck is solving the $\ell_p$ regression problem on the final subsampled matrix. The only remaining procedure is rounding and truncating the coordinates of $\mathbf{b} \in \mathbb{R}^n$ to form a vector $\mathbf{b}'' \in \mathbb{R}^n$ using the procedure RoundTrunc and then forming the groups $G_1, \ldots, G_\ell$, which clearly takes $\mathcal{O}(n)$ arithmetic operations combined. We thus analyze each of the three main runtime bottlenecks.

First observe that the extended matrix $\mathbf{M}$ is a Vandermonde matrix with $\mathcal{O}(dp^2)$ columns. (Cohen & Peng, 2015) show that $\mathcal{O}(\log n)$ matrix-vector multiplication operations can be done to compute approximate Lewis weights for the purposes of $\ell_p$ Lewis weight sampling. By Theorem A.4, each matrix-vector multiplication uses time $\mathcal{O}(n \log^2 n)$. Hence computing the extended matrix $\mathbf{M}$ uses $\mathcal{O}(n \log^3 n)$ time. Similarly, the extended matrix for each group $G_k$ is the product of a Vandermonde matrix with $\mathcal{O}(dp^2)$ columns and a diagonal matrix. Thus by Theorem A.4, the extended matrices for all the groups $G_k$ can be formed using $\mathcal{O}(n \log^3 n)$ time in total.

Since each Vandermonde matrix has $\mathcal{O}(dp^2)$ rows, then observe that each group $G_k$ samples $\mathcal{O}\left(\frac{dp^2}{\varepsilon^2} \log d\right)$ rows and there are $\ell = \mathcal{O}\left(\frac{1}{\varepsilon} \log n\right)$ such groups $k \in [\ell]$. Thus the resulting subsampled matrix has $\mathcal{O}\left(\frac{dp^2}{\varepsilon^3} \log^2 n\right)$ rows for $d \leq n$. To approximately solve the $\ell_p$ regression problem, Theorem A.5 notes that for $p \geq 2$ and a subsampled matrix of size $\mathcal{O}\left(\frac{dp^2}{\varepsilon^3} \log^2 d\right)$, we require only $\mathcal{O}\left(\frac{p\sqrt{d}}{\varepsilon^{3/2}} \log n \log \frac{d}{\varepsilon}\right)$ calls to a linear system solver. Moreover, on an iteration $t$ of the $\ell_p$ regression algorithm of (Adil et al., 2019a) used in Theorem A.5, the linear system solves the equation $\mathbf{x}_t \leftarrow (\mathbf{A}^\top \mathbf{R}_T \mathbf{A})^{-1} \mathbf{A}^\top \mathbf{R}_t \mathbf{b}$ for a diagonal matrix $\mathbf{R}_T$. Each linear system solve can be done in $d^\omega \operatorname{poly}\left(\frac{1}{\varepsilon}, p, \log n\right)$ time. Hence, the total time to approximately solve the $\ell_p$ regression problem is $d^{0.5+\omega} \operatorname{poly}\left(\frac{1}{\varepsilon}, p, \log n\right)$. Therefore, the total runtime is $\mathcal{O}(n \log^3 n) + d^{0.5+\omega} \operatorname{poly}\left(\frac{1}{\varepsilon}, p, \log n\right)$. $\square$

**Lemma 2.4.** *Let* $\mathbf{x} \in \mathbb{R}^n$ *and* $p = \Omega\left(\frac{\log n}{\varepsilon}\right)$. *Then* $\|\mathbf{x}\|_\infty \le \|\mathbf{x}\|_p \le (1+\varepsilon)\|\mathbf{x}\|_\infty$.

*Proof.* For any vector $\mathbf{x} \in \mathbb{R}^d$, we have

$$\|\mathbf{x}\|_\infty = \max_{i \in n} |x_i| \le \left(\sum_{i \in} |x_i|^p\right)^{1/p} \le n^{1/p} \cdot \max_{i \in n} |x_i|.$$

Since $(1+\varepsilon)^{3/eps} > e$ for all $\varepsilon > 0$, then $(1+\varepsilon)^p > n$ for $p = \Omega\left(\frac{\log n}{\varepsilon}\right)$. Therefore,

$$n^{1/p} \cdot \max_{i \in n} |x_i| \le (1+\varepsilon) \max_{i \in n} |x_i| = (1+\varepsilon)\|\mathbf{x}\|_\infty.$$

$\square$

**Corollary 2.5.** *Given* $\varepsilon \in (0,1)$ *and* $p \ge 1$, $\mathbf{A} \in \mathbb{R}^{n \times dq}$, *and* $\mathbf{b} \in \mathbb{R}^d$, *suppose* $\mathbf{A} = [\mathbf{A}_1| \dots |\mathbf{A}_q]$ *for Vandermonde matrices* $\mathbf{A}_1, \dots, \mathbf{A}_q \in \mathbb{R}^{n \times d}$. *Then there exists an algorithm that, with high probability, returns a vector* $\widehat{\mathbf{x}} \in \mathbb{R}^d$ *such that*

$$\|\mathbf{A}\widehat{\mathbf{x}} - \mathbf{b}\|_p \le (1+\varepsilon) \min_{\mathbf{x} \in \mathbb{R}^d} \|\mathbf{A}\mathbf{x} - \mathbf{b}\|_p,$$

*using* $\mathcal{O}\left(T(\mathbf{A})(dp)^{q-1}\log n + \text{poly}((dp)^q \log n, 1/\varepsilon)\right)$ *time, where* $T(\mathbf{A})$ *is the runtime of multiplying the matrix* $\mathbf{A}$ *by an arbitrary vector. For* $q = 2$, *this can be further optimized to* $\mathcal{O}\left(nd^{\omega_2/2-1} + \text{poly}((dp)^2, \log n, 1/\varepsilon)\right)$ *time, where* $\mathcal{O}(n^{\omega_2})$ *is the time to multiply an* $n \times n$ *matrix with an* $n \times n^2$ *matrix, so that* $\omega_2 \in [3,4]$.

*Proof.* Recall that a key part in the proof of Theorem 1.1 was to first the $\ell_{2^r}$ regression on a Vandermonde matrix with dimension $\mathbb{R}^{n \times d}$ to $\ell_1$ regression on a Vandermonde matrix with dimension $\mathbb{R}^{n \times (2^r(d-1)+1)}$, where $2^r \le p < 2^{r+1}$. For a matrix $\mathbf{A}$ with block Vandermonde structure, we can similarly write

$$(\langle \mathbf{a}_i, \mathbf{x} \rangle)^{2^r} = \left(\sum_{j=1}^{dq} a_{i,j} x_j\right)^{2^r} = \left(\sum_{k=1}^{q} \sum_{j=1}^{d} a_{i,2+(k-1)d}^{j-1} x_{j+(k-1)d}\right)^{2^r}$$

$$= \sum_{j=1}^{(2^r(d-1))^q+1} \prod_{k \in [q], \sum p_{j,k} = 2^r}^{q} a_{i,2+(k-1)d}^{p_{j,k}} y_j,$$

where again each $y_j$ is a fixed function of the coordinates of $\mathbf{x}$. Thus we can reshape the $\ell_{2^r}$ regression problem on a matrix $\mathbf{A}$ with dimension $\mathbb{R}^{n \times dq}$ with block Vandermonde structure to an $\ell_1$ regression problem on a matrix $\tilde{\mathbf{A}}$ with dimension $\mathbb{R}^{n \times (2^r(d-1))^q+1}$. Moreover, we can further reshape $\tilde{\mathbf{A}}$ into the concatenation of $(dp)^{q-1}$ Vandermonde matrices, where each Vandermonde matrix has columns that are geometrically growing in $a_{i,2}$ but are multiplied by all $(dp)^{q-1}$ products $\prod_{k=1}^{q-1} a_{i,2+kd}^{p_k}$, where $p_k \in [dp]$.

We can now use matrix-vector multiplication on each of the $(dp)^{q-1}$ Vandermonde matrices. Thus by Theorem A.2, we can $\ell_1$ Lewis weight sample from the rows of the reshaped $\tilde{\mathbf{A}}$, using $\mathcal{O}\left(T(\tilde{\mathbf{A}})(dp)^{q-1}\log n + (dp)^{\omega q}\right)$ time. We can similarly write $(\langle \mathbf{a}_i, \mathbf{x} - t_k \rangle)^{2^r}$ for each $t_k$ among the discretized values of the updated $\mathbf{b}$ vector as a sum of $(2^r(d-1))^q + 1$ terms that are all products of powers of the bases $a_{i,2}, a_{i,2+d}, \dots$ and a variables $y_j$, as in the proof of Theorem 1.1. Thus we can partition the $\ell_{2^r}$ regression problem into $\ell = \mathcal{O}\left(\frac{\log n}{\varepsilon}\right)$ instances of a constrained $\ell_1$ regression problem on $(dp)^{q-1}$ Vandermonde matrices, each with at most $2^r(d-1))^q + 1$ columns. To approximately solve the $\ell_p$ regression problem, we can thus sample rows by their $\ell_{p/2^r}$ Lewis weights, as in Theorem 1.1. Since there are up to $(dp)^{q-1}$ Vandermonde matrices, each with at most $2^r(d-1))^q + 1$ columns, then by Theorem A.2, the total time required is $\mathcal{O}\left(T(\mathbf{A})(dp)^{q-1}\log n + (dp)^{\omega q} \cdot \text{poly}(\log n, 1/\varepsilon)\right)$. $\square$

We remark that for the special case of $q = 2$, (Sa et al., 2018) noted an efficient bivariate matrix multiplication algorithm of (Nüsken & Ziegler, 2004; Kedlaya & Umans, 2011).

**Theorem A.7.** *(Nüsken & Ziegler, 2004; Kedlaya & Umans, 2011; Sa et al., 2018) Given a q-variate polynomial $f(X_1, \ldots, X_q)$ such that each variable has degree at most $d - 1$ and $N = d^q$ distinct points $x(i) = (x(i)_1, \ldots, x(i)_q)$ for $i \in [N]$, there exists an algorithm that uses $\mathcal{O}\left(d^{\omega_2(q-1)/2+1}\right)$ time to output the vector $(f(x(1)), \ldots, f(x(N)))$, where $\mathcal{O}\left(n^{\omega_2}\right)$ is the time to multiply an $n \times n$ matrix with an $n \times n^2$ matrix, so that $\omega_2 \in [3, 4]$.*

The case of a matrix-vector product for $q = 2$ corresponds to the evaluation of $d^2$ points in Theorem A.7. Thus we need to repeat the algorithm in Theorem A.7 a total of $\frac{n}{d^2}$ times to handle all $n$ rows in the input matrix. Since each instance of the algorithm uses $\mathcal{O}\left(d^{w_2/2+1}\right)$ time, the total time for the matrix-vector product is $\mathcal{O}\left(nd^{\omega_2/2-1}\right)$, rather than the naïve $\mathcal{O}\left(nd\right)$ time (recall that $\omega_2 \in [3, 4]$).

**Corollary A.8.** *Given $\varepsilon \in (0, 1)$ and $p \geq 1$, $\mathbf{A} \in \mathbb{R}^{n \times 2d}$, and $\mathbf{b} \in \mathbb{R}^d$, suppose $\mathbf{A} = [\mathbf{A}_1 | \mathbf{A}_2]$ for Vandermonde matrices $\mathbf{A}_1, \mathbf{A}_2 \in \mathbb{R}^{n \times d}$. Then there exists an algorithm that with high probability returns a vector $\widehat{\mathbf{x}} \in \mathbb{R}^d$ such that*

$$\|\mathbf{A}\widehat{\mathbf{x}} - \mathbf{b}\|_p \leq (1 + \varepsilon) \min_{\mathbf{x} \in \mathbb{R}^d} \|\mathbf{A}\mathbf{x} - \mathbf{b}\|_p,$$

*using $\mathcal{O}\left(nd^{\omega_2/2-1} + \text{poly}((dp)^2, \log n, 1/\varepsilon)\right)$, where $\mathcal{O}\left(n^{\omega_2}\right)$ is the time to multiply an $n \times n$ matrix with an $n \times n^2$ matrix.*

## A.2 $\ell_p$ REGRESSION ON OTHER STRUCTURED INPUT

---

**Algorithm 4** Faster regression for noisy low-rank matrices

---

**Input:** Rank $k$ matrix $\mathbf{K} \in \mathbb{R}^{n \times d}$, matrix $\mathbf{S} \in \mathbb{R}^{n \times d}$ with at most $s$ non-zero entries per row, such that $\mathbf{A} = \mathbf{K} + \mathbf{S}$, measurement vector $\mathbf{b}$, accuracy parameter $\varepsilon > 0$
**Output:** $\widehat{\mathbf{x}} \in \mathbb{R}$ with $\|\mathbf{A}\widehat{\mathbf{x}} - \mathbf{b}\|_p \leq (1 + \mathcal{O}(\varepsilon)) \min_{\mathbf{x} \in \mathbb{R}^d} \|\mathbf{A}\mathbf{x} - \mathbf{b}\|_p$
 1: $r \leftarrow \lfloor \log p \rfloor$                                                            $\triangleright 2^r \leq p < 2^{r+1}$.
 2: Extend $\mathbf{A}$ to a matrix $\mathbf{M}$ with dimension $n \times \mathcal{O}\left(d^s(k + s)^{2^r}\right)$ so that each entry $M_{i,j}$ is the coefficient of the $j$-th term in the tensor decomposition of $\mathbf{a}_i^{\otimes 2^r}$.
 3: Use $\ell_{p/2^r}$-Lewis weight sampling on $\mathbf{M}$ to find a set $S$ of $\mathcal{O}(d' \log d')$ indices in $[n]$ and rescaling factors, for $d' = \mathcal{O}\left(d^s(k + s)^{2^r}\right)$.
 4: Let $\mathbf{A}'$ be the corresponding submatrix of $\mathbf{A}$ with indices in $S$ and scaled accordingly.
 5: Compute $\tilde{\mathbf{x}} \leq 5 \min_{\mathbf{x} \in \mathbb{R}^d} \|\mathbf{A}'\mathbf{x} - \mathbf{b}\|_p$.
 6: $\mathbf{b}' \leftarrow \mathbf{b} - \mathbf{A}\tilde{\mathbf{x}}$, $\mathbf{b}'' \leftarrow \mathsf{RoundTrunc}(\mathbf{b}')$, $\ell \leftarrow \mathcal{O}\left(\frac{\log n}{\varepsilon}\right)$
 7: Partition the rows of $\mathbf{A}$ into groups $G_1, \ldots, G_\ell$, each containing all rows with the same value of $\mathbf{b}''$
 8: Let $\mathbf{G}_k$ be the corresponding submatrix and $t_k$ be the coordinate of $\mathbf{b}''$ corresponding to $G_k$ for each $k \in [\ell]$.
 9: Use $\ell_{p/2^r}$-Lewis weight sampling on $[\mathbf{G}_k; t_k]$ to find a set $S'_k$ of $\mathcal{O}\left(\frac{d''}{\varepsilon^2} \log d''\right)$ indices in $[n]$ and rescaling factors, where $d'' = \mathcal{O}\left(pd^s(k + s)^{2^r}\right)$.
10: Let $\mathbf{T}_k$ be the corresponding sampling and rescaling matrix for $S'_k$.
11: $\mathbf{T} \leftarrow [\mathbf{T}_1; \ldots; \mathbf{T}_k]^\top$
12: Compute $\widehat{\mathbf{x}} \leq (1 + \varepsilon) \min_{\mathbf{x} \in \mathbb{R}^d} \|\mathbf{T}\mathbf{A}\mathbf{x} - \mathbf{T}\mathbf{b}\|_p$.
13: **return** $\widehat{\mathbf{x}}$

---

**Lemma A.9.** *Given $\varepsilon \in (0, 1)$ and $p \geq 1$, a matrix $\mathbf{A} \in \mathbb{R}^{n \times d}$ such that $\mathbf{A} = \mathbf{K} + \mathbf{S}$ for a rank $k$ matrix $\mathbf{K}$ and an $s$-sparse matrix $\mathbf{S}$, and $\mathbf{b} \in \mathbb{R}^d$, there exists an algorithm that with high probability, returns a vector $\widehat{\mathbf{x}} \in \mathbb{R}^d$ such that*

$$\|\mathbf{A}\widehat{\mathbf{x}} - \mathbf{b}\|_p \leq (1 + \varepsilon) \min_{\mathbf{x} \in \mathbb{R}^d} \|\mathbf{A}\mathbf{x} - \mathbf{b}\|_p.$$

*Proof.* The proof is similar to Lemma A.3. We once again let $r$ be an integer so that $2^r \le p < 2^{r+1}$ and observe that

$$\sum_{i \in [n]} |\langle \mathbf{a}_i, \mathbf{x} \rangle|^p = \sum_{i \in [n]} |(\langle \mathbf{a}_i, \mathbf{x} \rangle)^{2^r}|^{p/2^r} = \sum_{i \in [n]} |(\langle \mathbf{k}_i + \mathbf{s}_i, \mathbf{x} \rangle)^{2^r}|^{p/2^r}.$$

Since $\mathbf{K}$ is a low-rank matrix, then for all $\mathbf{k}_i$, we can write

$$\mathbf{k}_i = \sum_{j=1}^{k} \alpha_{i,j} \mathbf{v}_j,$$

for a fixed set of basis vectors $\mathbf{v}_1, \dots, \mathbf{v}_k \in \mathbb{R}^d$. Hence we have

$$\sum_{i \in [n]} |\langle \mathbf{a}_i, \mathbf{x} \rangle|^p = \sum_{i \in [n]} \left| \left\langle \mathbf{s}_i + \sum_{j=1}^{k} \alpha_{i,j} \mathbf{v}_j, \mathbf{x} \right\rangle^{2^r} \right|^{p/2^r}.$$

Since $\mathbf{S}$ has sparsity $s$, then we can further write

$$\sum_{i \in [n]} |\langle \mathbf{a}_i, \mathbf{x} \rangle|^p = \sum_{i \in [n]} \left| \left\langle \sum_{j=1}^{s} \beta_{i,j} \mathbf{e}_{i_j} + \sum_{j=1}^{k} \alpha_{i,j} \mathbf{v}_j, \mathbf{x} \right\rangle^{2^r} \right|^{p/2^r},$$

where $\mathbf{e}_1, \dots, \mathbf{e}_d$ denote the elementary vectors. By the Hadamard Product-Kronecker Product mixed-product property, we have

$$\sum_{i \in [n]} |\langle \mathbf{a}_i, \mathbf{x} \rangle|^p = \sum_{i \in [n]} \left| (\mathbf{y}_1 \otimes \dots \mathbf{y}_{2^r}) \odot \mathbf{x}^{\otimes(2^r)} \right|^{p/2^r},$$

where each $\mathbf{y}_k \in \{\alpha_{i,1}\mathbf{v}_1, \dots, \alpha_{i,k}\mathbf{v}_k, \beta_{i,1}\mathbf{e}_{i_1}, \dots, \beta_{i,s}\mathbf{e}_{i_s}\}$ for $i \in [2^r]$. Thus for a fixed set of elementary vectors $\mathbf{e}_{i_1}, \dots, \mathbf{e}_{i_s}$, there are $(k+s)^{2^r}$ possible values for the tensor product $(\mathbf{y}_1 \otimes \dots \mathbf{y}_{2^r})$. Since there are $\binom{d}{s}$ choices for the elementary vectors $\mathbf{e}_{i_1}, \dots, \mathbf{e}_{i_s}$, then there are at most $(Cd^s(k+s)^{2^r})$ possible values for the tensor product for an absolute constant $C > 0$. Therefore, the $\ell_p$ subspace embedding problem on $\mathbf{A} \in \mathbb{R}^{n \times d}$ can be reshaped as a constrained $\ell_{p/2^r}$ subspace embedding problem on an input matrix of size $n \times (Cd^s(k+s)^{2^r})$. Hence for $\ell_p$ regression with $p \in [2^r, 2^{r+1})$, we have

$$\sum_{i \in [n]} |\langle \mathbf{a}_i, \mathbf{x} \rangle|^p = \sum_{i \in [n]} \left| \sum_{j=1}^{2^r(d-1)+1} M_{i,j} y_j \right|^{p/2^r},$$

which is a constrained $\ell_{p/2^r}$ regression problem on a matrix $\mathbf{M}$ of size $n \times (Cd^s(k+s)^{2^r})$ whose entries can be determined from the decomposition of each row of $\mathbf{A}$.

Using $\ell_{p/2^r}$ Lewis weight sampling, Theorem A.2 implies that we can find a matrix $\mathbf{M}'$ such that

$$\frac{11}{12} \|\mathbf{M}\mathbf{y}\|_{p/2^r}^{p/2^r} \le \|\mathbf{M}'\mathbf{y}\|_{p/2^r}^{p/2^r} \le \frac{13}{12} \|\mathbf{M}\mathbf{y}\|_{p/2^r}^{p/2^r}$$

for all $\mathbf{y} \in \mathbb{R}^{(Cd^s(k+s)^{2^r})}$ with high probability. By Lemma A.1, we can thus compute a vector $\tilde{\mathbf{x}}$ such that

$$\|\mathbf{A}\tilde{\mathbf{x}} - \mathbf{b}\|_p \le 12 \min_{\mathbf{x} \in \mathbb{R}^d} \|\mathbf{A}\mathbf{x} - \mathbf{b}\|_p.$$

We again set $\mathbf{b}' = \mathbf{b} - \mathbf{A}\tilde{\mathbf{x}}$ and define $\text{OPT} = \min_{\mathbf{x} \in \mathbb{R}^d} \|\mathbf{A}\mathbf{x} - \mathbf{b}\|_p$, so that

$$\|\mathbf{b}'\|_p = \|\mathbf{A}\tilde{\mathbf{x}} - \mathbf{b}\|_p \le 12\text{OPT}.$$

Let $\mathbf{b}' = \mathbf{b} - \mathbf{A}\tilde{\mathbf{x}}$ and $\text{OPT} = \min_{\mathbf{x} \in \mathbb{R}^d} \|\mathbf{A}\mathbf{x} - \mathbf{b}\|_p$, so that

$$\|\mathbf{b}'\|_p = \|\mathbf{A}\tilde{\mathbf{x}} - \mathbf{b}\|_p \le 12\text{OPT}.$$

Let $\mathbf{b}''$ be the vector with all entries of $\mathbf{b}'$ rounded to the nearest power of $(1 + \varepsilon)$, starting at the maximum entry of $\mathbf{b}'$ in absolute value, and stopping after we are $\frac{1}{\text{poly}(n)}$ of that and replacing all remaining entries with 0. By the triangle inequality, we have

$$\|\mathbf{A}\mathbf{x} - \mathbf{b}''\|_p \leq \|\mathbf{A}\mathbf{x} - \mathbf{b}'\|_p + \|\mathbf{b}' - \mathbf{b}''\|_p \leq \|\mathbf{A}\mathbf{x} - \mathbf{b}'\|_p + 12\varepsilon\text{OPT} \leq (1 + 12\varepsilon)\|\mathbf{A}\mathbf{x} - \mathbf{b}''\|_p,$$

for any $\mathbf{x} \in \mathbb{R}^{d+1}$.

Since the coordinates of $\mathbf{b}''$ can have $\ell = \mathcal{O}\left(\frac{\log n}{\varepsilon}\right)$ possible distinct values, we can partition the rows of $\mathbf{A}$ into $\ell$ groups, $G_1, \ldots, G_\ell$, based on the corresponding values of $\mathbf{b}''$. Let $t_m$ be the corresponding value of $\mathbf{b}''$ for all rows in a group $G_m$, so that

$$\|\mathbf{A}\mathbf{x} - \mathbf{b}''\|_p^p = \sum_m \sum_{i \in G_k} |\langle \mathbf{a}_i, \mathbf{x} \rangle - t_m|^p.$$

By the above argument, we have for each $i \in G_k$,

$$(\langle \mathbf{a}_i, \mathbf{x} \rangle - t_m)^{2^r} = \left| -t_m + \left\langle \sum_{j=1}^{s} \beta_{i,j} \mathbf{e}_{i_j} + \sum_{j=1}^{k} \alpha_{i,j} \mathbf{v}_j, \mathbf{x} \right\rangle \right|^{2^r} = \sum_{j=1}^{2^r C d^s (k+s)^{2^r}} B_{i,j} t_{m,j} y_j,$$

where (1) $B_{i,j}$ are entries of a matrix $\mathbf{B}$ with $2^r C d^s (k+s)^{2^r}$ columns that can be computed from $\mathbf{A}$, (2) $t_{m,1}, t_{m,2}, \ldots$ are fixed values that can be computed from $m_k$, and (3) each $y_j$ is a fixed function of the coordinates of $\mathbf{x}$. Notably, $\mathbf{B}$ is the matrix formed by the concatenation of the coefficients of the decomposition of the $\alpha$-fold tensor product of the row into the $\alpha$-fold tensor products of the low-rank and sparse basis elements, for each $\alpha = 0, \ldots, p$. By comparison, the matrix $\mathbf{M}$ previously defined in this proof is only the decomposition for $\alpha = p$. Hence, the $\ell_{2^r}$ regression problem on a submatrix of $\mathbf{A} \in \mathbb{R}^{n \times d}$ the same coordinate of $\mathbf{b}''$, can be reshaped as a constrained $\ell_1$ regression problem on a matrix with $2^r C d^s (k+s)^{2^r}$ columns.

The remainder of the proof follows from the same grouping argument as Theorem 1.1. We apply Theorem A.2 by sampling rows of $\mathbf{B}$ corresponding to their $\ell_{p/2^r}$ Lewis weights in the submatrix induced by the rows of $G_m$, we obtain a matrix $\mathbf{T}_m$ such that with high probability,

$$(1-\varepsilon)\|\mathbf{T}_m \mathbf{B}\mathbf{y} - \mathbf{T}_m \mathbf{v}_m\|_{p/2^r}^{p/2^r} \leq \sum_{i \in G_m} \left( \sum_{j=1}^{2^r C d^s (k+s)^{2^r}} B_{i,j} t_{m,j} y_j \right)^{p/2^r} \leq (1+\varepsilon)\|\mathbf{T}_m \mathbf{B}\mathbf{y} - \mathbf{T}_m \mathbf{v}_m\|_{p/2^r}^{p/2^r},$$

where $\mathbf{v}_m$ is the vector restricted to the coordinates of $\mathbf{b}''$ that are equal to $t_m$. Conditioning on the above inequality holding, it follows that

$$(1 - \varepsilon)\|\mathbf{T}_m \mathbf{A}\mathbf{x} - \mathbf{T}_m \mathbf{v}_m\|_p^p \leq \sum_{i \in G_m} |\langle \mathbf{a}_i, \mathbf{x} \rangle - t_m|^p \leq (1 + \varepsilon)\|\mathbf{T}_m \mathbf{A}\mathbf{x} - \mathbf{T}_m \mathbf{v}_m\|_p^p.$$

Therefore by summing over all $m \in [\ell]$, we have that with high probability,

$$(1 - \varepsilon)\sum_m \|\mathbf{T}_m \mathbf{A}\mathbf{x} - \mathbf{T}_m \mathbf{v}_m\|_p^p \leq \sum_m \sum_{i \in G_m} |\langle \mathbf{a}_i, \mathbf{x} \rangle - t_m|^p \leq (1 + \varepsilon)\sum_m \|\mathbf{T}_m \mathbf{A}\mathbf{x} - \mathbf{T}_m \mathbf{v}_m\|_p^p.$$

For $\mathbf{T} = \mathbf{T}_1 \circ \ldots \circ \mathbf{T}_\ell$, we have $\sum_m \|\mathbf{T}_m \mathbf{A}\mathbf{x} - \mathbf{T}_m \mathbf{v}_m\|_p^p = \|\mathbf{T}\mathbf{x} - \mathbf{b}''\|_p^p$. Since $\|\mathbf{A}\mathbf{x} - \mathbf{b}''\|_p^p = \sum_m \sum_{i \in G_m} |\langle \mathbf{a}_i, \mathbf{x} \rangle - t_m|^p$, then

$$(1 - \varepsilon)\|\mathbf{T}\mathbf{A}\mathbf{x} - \mathbf{T}\mathbf{b}''\|_p^p \leq \|\mathbf{A}\mathbf{x} - \mathbf{b}''\|_p^p \leq (1 + \varepsilon)\|\mathbf{T}\mathbf{A}\mathbf{x} - \mathbf{T}\mathbf{b}''\|_p^p.$$

Therefore for $p \geq 1$,

$$(1 - \varepsilon)\|\mathbf{T}\mathbf{A}\mathbf{x} - \mathbf{T}\mathbf{b}''\|_p \leq \|\mathbf{A}\mathbf{x} - \mathbf{b}''\|_p \leq (1 + \varepsilon)\|\mathbf{T}\mathbf{A}\mathbf{x} - \mathbf{T}\mathbf{b}''\|_p.$$

Because $\ell = \mathcal{O}\left(\frac{\log n}{\varepsilon}\right)$ and $\sum \mathcal{O}\left(T(G_k)\right) = \mathcal{O}\left(T(\mathbf{A})\right)$, we can compute a vector $\widehat{\mathbf{x}} \in \mathbb{R}^d$ such that

$$\|\mathbf{A}\widehat{\mathbf{x}} - \mathbf{b}\|_p \leq (1 + \mathcal{O}(\varepsilon)) \min_{\mathbf{x} \in \mathbb{R}^d} \|\mathbf{A}\mathbf{x} - \mathbf{b}\|_p.$$

$\square$

**Lemma A.10.** *Given the low-rank factorization of* $\mathbf{K}$*,* *Algorithm 4 uses* $n \operatorname{poly}\left(2^p, d^s, k^p, s^p, \frac{1}{\varepsilon}, \log n\right)$ *time.*

*Proof.* We analyze the runtime of Algorithm 4. First note that we can compute the extended matrix in time $\mathcal{O}\left(n((k+1)^p d^s s^p)\right)$ to perform the $\ell_{p/2^r}$ Lewis weight sampling, where we recall that $r$ is the unique integer such that $2^r \leq p < 2^{r+1}$. To perform Lewis weight sampling on the extended matrix, we require matrix-vector multiplication, which requires time $\mathcal{O}(nk)$ for a low-rank matrix and time $\mathcal{O}(ns)$ for a matrix whose rows have at most $s$ nonzero entries.

After the first iteration of Lewis weight sampling, we can round and truncate the coordinates of $\mathbf{b} \in \mathbb{R}^n$ to form a vector $\mathbf{b}'' \in \mathbb{R}^n$ using the procedure RoundTrunc and then forming the groups $G_1, \ldots, G_\ell$, which clearly takes $\mathcal{O}(n)$ arithmetic operations combined. Once the groups are formed, we can compute the extended matrix in time $\mathcal{O}\left(n((k+1)^p d^s s^p)\right)$ and perform $\ell_{p/2^r}$ Lewis weight sampling on each group, which takes $\mathcal{O}(nk + ns)$ total time across all groups. To approximately solve the resulting $\ell_p$ regression problem formed by the subsampled rows, we require $\operatorname{poly}\left(d^s, k^p, s^p, \frac{1}{\varepsilon}\right)$ time. Therefore, the total runtime is $n \operatorname{poly}\left(2^p, d^s, k^p, s^p, \frac{1}{\varepsilon}, \log n\right)$. $\square$

Theorem 1.5 then follows from Lemma A.9 and Lemma A.10. Theorem 1.6 is achieved through similar analysis for a noisy Vandermonde matrix. In particular, it follows from the same proof structure as Lemma A.3 by showing for $2^r \leq p < 2^{r+1}$, the $\ell_{2^r}$ subspace embedding problem on $\mathbf{A} = \mathbf{V} + \mathbf{S}$ for a given Vandermonde matrix $\mathbf{V} \in \mathbb{R}^{n \times d}$ and a sparse matrix $\mathbf{S} \in \mathbb{R}^{n \times d}$ can be reshaped as a constrained $\ell_1$ subspace embedding problem on an input Vandermonde matrix of size $n \times d'$, where $d' = (d^s)(s^p)(pd)$.

## B  APPLICATIONS TO POLYNOMIAL REGRESSION

In the polynomial regression problem, the goal is to find a degree $d$ polynomial $\hat{q}$ such that

$$\|\widehat{q}(t) - f(t)\|_p^p \leq (1+\varepsilon) \cdot \min_{q:\deg(q) \leq d} \|q(t) - f(t)\|_p^p,$$

where $\varepsilon > 0$ is an accuracy parameter given as input and $\|\cdot\|_p^p$ is the $\ell_p$ norm to the $p^{\text{th}}$ power, $\|g\|_p^p = \int_{-1}^{1} |g(t)|^p dt$. The polynomial regression problem is a fundamental problem in statistics, computational mathematics, machine learning, and more. The problem has been studied as early as the 19th century with the work of Legendre and Gauss on least squares polynomial regression and has applications in learning half-spaces (Kalai et al., 2008), solving parametric PDEs (Hampton & Doostan, 2015), and surface reconstruction (Pratt, 1987).

Given the flexibility to choose query locations $x_1, \ldots, x_s$, we can consider the polynomial regression problem as an *active learning* or *experimental design* problem. Thus we would like to minimize the number of queries $s$, as a function of the approximation degree $d$, the norm $p$, and the accuracy parameter $\varepsilon$, to find $\hat{q}$. Observe that $d+1$ queries are obviously necessary, but also that $d+1$ queries suffice when $f$ can be exactly fit by a degree $d$ polynomial, by using direct interpolation. In general, however, in the case when $\min_{q:\deg(q) \leq d} \|q(t) - f(t)\|_p^p \neq 0$ we require $s > d + 1$ queries. Our Vandermonde $\ell_p$ regression results can be used to give the first result showing that for all $p \geq 1$, $d \exp(O(p)) \cdot \operatorname{poly}\left(\frac{1}{\varepsilon}\right)$ queries suffice to obtain a $(1+\varepsilon)$-approximation to the best polynomial fit.

We require the following structural theorem reducing the $\ell_p$ polynomial regression problem to a problem of solving $\ell_p$ regression on Vandermonde matrices:

**Theorem B.1.** *(Kane et al., 2017; Meyer et al., 2021) Suppose $s_1, \ldots, s_{n_0}$ are drawn uniformly from $[-1, 1]$. Let $\mathbf{A} \in \mathbb{R}^{n_0 \times (d+1)}$ be the associated Vandermonde matrix, so that $\mathbf{A}_{i,j} = s_i^{j-1}$. Let $n_0 = \exp(O(p))\tilde{O}\left(\frac{1}{\varepsilon^{2+2p}} d^5\right)$ and let $\mathbf{b} \in \mathbb{R}^{n_0}$ be the evaluations of $f$, so that $\mathbf{b}_i = f(s_i)$. Then with probability $\frac{11}{12}$, the sketched solution $\hat{\mathbf{x}} = \operatorname{argmin}_{\mathbf{x}} \|\mathbf{A}\mathbf{x} - \mathbf{b}\|_p$ satisfies*

$$\|\mathcal{P}\mathbf{x} - f\|_p^p \leq (1+\varepsilon) \min_{\mathbf{x} \in \mathbb{R}^{d+1}} \|\mathcal{P}\mathbf{x} - f\|_p^p.$$

Theorem B.1 states that we can uniformly sample $\exp(O(p)) \cdot \operatorname{poly}\left(d, \frac{1}{\varepsilon}\right)$ points from $[-1, 1]$. We can then form an $\ell_p$ regression problem by using the evaluation each of the polynomial bases

at the sampled points to form the design matrix $\mathbf{A}$ and querying the underlying signal $f$ at the sampled points to form the measurement vector $\mathbf{b}$. Theorem B.1 says that the optimal solution $\hat{\mathbf{x}} = \operatorname{argmin}_{\mathbf{x}} \|\mathbf{A}\mathbf{x} - \mathbf{b}\|_p$ is a $(1 + \varepsilon)$-approximation to the best fit degree $d$ polynomial. We can naïvely approximately solve the $\ell_p$ regression on the Vandermonde matrix $\mathbf{A}$ and the measurement vector $\mathbf{b}$ by standard $\ell_p$ regression techniques such as Lewis weight sampling, which would result in total query complexity $\exp(O(p))\tilde{O}\left(\frac{1}{\varepsilon^{2+2p}} d^5\right)$. However, we can instead note that Lemma 1.4 implies we can instead solve an $\ell_q$ regression problem on a Vandermonde matrix with $O(dp)$ columns for $q \in [1, 2]$. Crucially for $q \in [1, 2]$, there exist active $\ell_q$ regression algorithms that only require reading $\tilde{O}(d) \cdot \operatorname{poly}\left(\frac{1}{\varepsilon}\right)$ entries of $\mathbf{b}$:

**Theorem B.2.** *(Musco et al., 2021) Given $p \in [1, 2]$ and an input matrix $\mathbf{A} \in \mathbb{R}^{n \times d}$, there exists an algorithm that reads $\tilde{O}(d) \cdot \operatorname{poly}\left(\frac{1}{\varepsilon}\right)$ entries of $\mathbf{b}$ and with probability at least $0.99$, outputs $\tilde{\mathbf{x}}$ such that*

$$\|\mathbf{A}\mathbf{x} - \mathbf{b}\|_p \le (1 + \varepsilon)\min_{\mathbf{x}} \|\mathbf{A}\mathbf{x} - \mathbf{b}\|_p.$$

Hence by Theorem B.2, we can approximately solve the $\ell_q$ regression on a Vandermonde matrix with $dp$ rows by reading $\tilde{O}(dp) \cdot \operatorname{poly}\left(\frac{1}{\varepsilon}\right)$ entries of $\mathbf{b}$. By Lemma 1.4, the approximate solution will also be a $(1 + \varepsilon)$-approximation to the optimal $\ell_p$ regression on the Vandermonde matrix $\mathbf{A}$. By Theorem B.1, the approximate solution will also form the coefficient vector of a polynomial that is a $(1 + \varepsilon)$-approximation to the polynomial $\ell_p$ regression problem. Therefore, we obtain the following guarantees for the polynomial regression problem:

**Theorem B.3.** *For any degree $d$ and norm $p \ge 1$, there exists an algorithm that queries $f$ at $s = dp \operatorname{poly}\left(\log(dp), \frac{1}{\varepsilon}\right)$ points and outputs a degree $d$ polynomial $\hat{q}(t)$ such that*

$$\|\hat{q}(t) - f(t)\|_p^p \le (1 + \varepsilon) \cdot \min_{q:\deg(q) \le d} \|q(t) - f(t)\|_p^p,$$

*with probability at least $\frac{2}{3}$.*

The previous-best algorithm (Kane et al., 2017; Meyer et al., 2021) for $\ell_p$ polynomial regression sampled $O(d^5)$ points uniform from $[-1, 1]$ and then used standard $\ell_p$ regression algorithms that required reading the signal at all $O(d^5)$ sampled points, for a total query complexity of $O(d^5)$. By comparison, Theorem B.3 only has linear dependency in $d$ due to the structural property of Lemma 1.4. Since $\Omega(d)$ queries are clearly necessary, our result settles the dependency of $d$ in the query complexity for $\ell_p$ polynomial regression for all $d$ and all $p \ge 1$.

## C  ADDITIONAL EXPERIMENTS

In this section, we experimentally verify that $\ell_{p/2^r}$ Lewis Weight sampling works for unstructured matrices. We take a similar approach as in Section 3 to verify that $\ell_{p/2^r}$ Lewis Weight sampling is correct for $\ell_p$ regression on unstructured matrices. For this test, we fix $n = 25,000$, $d = 10$, and $p = 6$, while varying $m \in [1, 1000]$. We let $\mathbf{A} = \begin{bmatrix} \mathbf{G}_1 & \mathbf{0} \\ \mathbf{0} & \mathbf{G}_2 \end{bmatrix}$ where $\mathbf{G}_1 \in \mathbb{R}^{100 \times 6}$ and $\mathbf{G}_2 \in \mathbb{R}^{24,900 \times 4}$ are i.i.d. $N(0, 1)$ matrices. To generate $\mathbf{b}$, we sample a vector $\mathbf{x} \in \mathbb{R}^{10}$ whose first 6 entries are $N(0, 100^2)$ and remaining 4 entries are $N(0, 1)$, and let $\mathbf{b} = \mathbf{A}\mathbf{x} + \mathbf{z}$ where $\mathbf{z} \in \mathbb{R}^{25,000}$ is a $N(0, 1)$ iid vector.

We generate this matrix $\mathbf{A}$ and response vector $\mathbf{b}$ just once and run $\ell_{p/2^r}$ Lewis Weight sampling many times, so the variance in the plot comes only from the random sampling algorithms. Note that we again omit the rounding procedure on $\mathbf{b}$. Figure 3 shows the result of this test, and we clearly see that the error shrinks quickly in $m$ for our algorithm. This approach is much more practical than the prior Lewis Weight approximation method for unstructured matrices when $p > 4$. That approach required solving a non-linearly constrained SDP $\mathcal{O}(\log n)$ times (Cohen & Peng, 2015), while our method requires only a Gaussian sketch matrix and the standard Lewis Weight Iteration, which converges very quickly.

Since $\mathbf{b}$ is so large on its first 100 entries, it is important for any subsampling algorithm to sample at least 6 of the first 100 rows. Uniform sampling picks none of these rows until $m \approx \frac{25000}{100} = 250$, which is why uniform sampling fails to converge to a good solution for small $m$. Lewis weight sampling instead gives much higher priority to the first 100 rows, avoiding any issue. This is why the the gap between Lewis Weight sampling and uniform sampling is so large for this experiment.

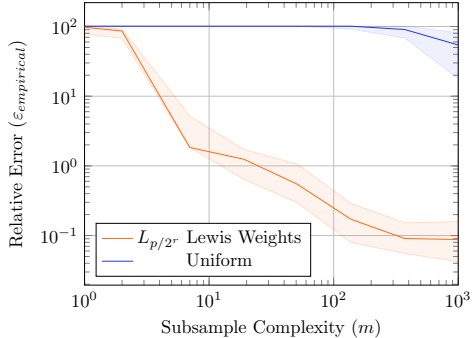

Fig. 3: Empirical Relative Error ($\varepsilon_{empirical}$) versus subsample complexity $m$. We ran the experiment 50 times and plot the median, $25^{th}$ quartile, and $75^{th}$ quartile for each value of $m$.

## D  ADDITIONAL RELATED WORK

As previously mentioned, subspace embeddings are common tools used to approximately solve $\ell_p$ regression. Given an input matrix $\mathbf{A} \in \mathbb{R}^{n \times d}$, a subspace embedding is a matrix $\mathbf{M} \in \mathbb{R}^{m \times d}$ with $m \ll n$ such that
$$(1 - \varepsilon)\|\mathbf{A}\mathbf{x}\|_p \leq \|\mathbf{M}\mathbf{x}\|_p \leq (1 + \varepsilon)\|\mathbf{A}\mathbf{x}\|_p,$$
for all $\mathbf{x} \in \mathbb{R}^d$. Thus given an instance of $\ell_p$ regression, where the goal is to minimize $\|\mathbf{A}\mathbf{x} - \mathbf{b}\|_p$, we can set $\mathbf{B} = [\mathbf{A}; \mathbf{b}]$, compute a subspace embedding for $\mathbf{B}$, and then solve a constrained $\ell_p$ regression problem on the smaller matrix $\mathbf{B}$.

A subspace embedding of a matrix $\mathbf{A} \in \mathbb{R}^{n \times d}$ can be formed by sampling rows of $\mathbf{A}$ with probabilities proportional to their $\ell_p$ leverage scores, e.g., (Cormode et al., 2018; Dasgupta et al., 2008) their $\ell_p$ sensitivities, e.g., (Clarkson et al., 2019; Braverman et al., 2020; 2021; Musco et al., 2021); or their $\ell_p$ Lewis weights, e.g., (Cohen & Peng, 2015; Durfee et al., 2018; Chhaya et al., 2020; Chen & Derezinski, 2021; Parulekar et al., 2021; Meyer et al., 2021). In any of these cases, the resulting matrix $\mathbf{M}$ will contain a subset of rows of $\mathbf{A}$ that are rescaled by a function of their sampling probability, as to give an unbiased estimate of the actual $\ell_p$ mass.

In addition to sampling methods, sketching is a common approach for subspace embeddings. In these cases, the subspace embedding $\mathbf{M}$ is formed by setting $\mathbf{M} = \mathbf{R}\mathbf{A}$ for some (often random) matrix $\mathbf{R} \in \mathbb{R}^{m \times n}$. The advantage of sketching over sampling is that sometimes $\mathbf{R}$ can be computed oblivious to the structure of $\mathbf{A}$, whereas the sampling probabilities for each of the above distributions ($\ell_p$ leverage scores, $\ell_p$ sensitivities, and $\ell_p$ Lewis weights) are data dependent and thus require a pass over the matrix $\mathbf{A}$. The sketching matrix can be generated from a family of random matrices whose entries are Cauchy random variables for $p = 1$, e.g., (Sohler & Woodruff, 2011; Meng & Mahoney, 2013; Clarkson et al., 2016), or sub-Gaussian random variables for $p = 2$, e.g., (Sarlós, 2006; Nelson & Nguyen, 2013; Clarkson & Woodruff, 2013). More generally, exponential random variables can be used for $p \geq 2$, though the number of rows $m$ in the resulting sketch matrix now has a polynomial dependency in $n$ (Woodruff & Zhang, 2013). A line of recent works has studied the tradeoffs between oblivious linear sketches, sampling-based algorithms, and other sketches for $\ell_p$ subspace embeddings (Wang & Woodruff, 2019; Li et al., 2021). In this paper, we focus on sampling-based algorithms for $\ell_p$ regression due to preservation of structure when the input matrix $\mathbf{A}$ itself has structure.

