# OpenReview forum: "Fast Regression for Structured Inputs"
_ICLR.cc/2022/Conference — ICLR 2022 Poster_

### Official Review · Reviewer_KU5L · 2021-10-19

**Correctness:** 4
**Technical Novelty And Significance:** 4
**Empirical Novelty And Significance:** 3
**Recommendation:** 10
**Confidence:** 4

**Main Review:**

I find the new algorithmic ideas quite interesting. Much of the recent work on randomized numerical linear algebra, especially those involving p-norms, deal with worst case matrices. It's quite surprising to me that using additional structural information, it's possible to perturb entries of the matrices to simplify the sampling processes.

A significant contribution (which I neglected in my initial read) is that the poly(p) * d bound shown here also work for the p > 2 case. To me this is highly surprising: all previous results require d^{p/2} rows, and such bound is tight for general matrices. This is very intriguing: it clearly demonstrates the value of the approach of looking at specialized matrix classes, and will likely motivate further investigations.

A subtle issue about the initial experiments is that the gains of biased sampling can only be demonstrated on highly non-uniform matrices. This has been addressed in the revision via additional experiments. Here the incoherence of Vandermonde matrices also provides further motivations for studying such instances.

**Summary Of The Paper:**

This paper studies ways of subsampling tall-and-dense p-norm regression involving structured Vandermonde matrices. It shows that in this setting with additional structure, sampling by Lewis weights produces poly(p) * d row sized samples for all values of p. Theoretically, this has two major advantages: it avoids more complicated computations of Lewis weights for higher values of p, and the sample size required is significantly better than the d^{p/2} bound for general matrices.

The algorithm is empirically evaluated on 25k-by-10 synthetic matrices, where it was observed that about 1k rows suffice to give sketches that incur at most 10% relative error. Furthermore, additional experiments provide cases where this type of sampling do significantly better than sampling uniformly.

**Summary Of The Review:**

This paper takes an interesting, practically well motivated, study of sketching p-norm regression. The connections with polynomial regression is quite powerful, and I'm only slowly getting a sense of its value.

Both its theoretical and practical results are quite surprising and interesting to those working on optimization/randomized numerical algorithms.

---

### Official Review · Reviewer_JQBd · 2021-11-02

**Correctness:** 4
**Technical Novelty And Significance:** 2
**Empirical Novelty And Significance:** Not applicable
**Recommendation:** 6
**Confidence:** 3

**Main Review:**

Strengths
1. Author gives algorithms that return samples whose sizes are significantly smaller than existing sample sizes for both Lp and L_\infty regression.
2. They define a very useful property called rank for regression, which is extensively used in their analysis.
3. The algorithms, theorems and lemmas are very well written.

Wekaness
1. Missing related work. Authors should discuss how subspace embedding has been used to get a quick approximation of Lp regression (Clarkson., et al. "The fast cauchy transform and faster robust linear regression.", Chhaya, et al. "Streaming coresets for symmetric tensor factorization.", Woodruff and Zhang. "Subspace embeddings and\ell_p-regression using exponential random variables." etc).
2. ||Ax-b||_p^p can be written as ||By||_p^p, for B = [A, b] (augmenting b as last column) and y = [x, -1] (setting last index as -1). Now by doing this the structure of B might slightly differ from the structure of A. So discussing the possibility of solving the actual problem by preserving the subspace of B would be a better start to motivate towards "Rounding and truncating measurement vector" (section 1.2). In particular, I am also curious that how much your guarantee improves by using the rounding and grouping technique instead of just relying on the structure of B (slightly differed from A).
3. In the abstract it was mentioned that you get a sublinear time algorithm for L_\infty regression but the running time of Theorem 1.2 is only O(n). Can you clarify, what do you mean by a sublinear time algorithm.
4. Weak empirical evaluation. The author must give compare the performance (relative error and running time) from their sampling technique with other naive sampling techniques such as uniform, standard lewis weights sampling, or using Cauchy random variables (Clarkson et. al).

More comment
In theorem 1.1 and 1.2 the \eps for S should be in (0,1/3) in order to get (1+\delta) approximation to the optimal solution where \delta \in (0,1).

Update:
I appreciate for giving a satisfactory reply. I am willing to raise my score by 1.

**Summary Of The Paper:**

The paper creates a coreset for solving Lp regression on structured input. For these cases, the paper shows that the coreset size is only polynomial in p. The author defines a notion of rank to Lp regression problem and uses a rounding and grouping technique on the response vector to improve their sample size. They also give a sublinear time algorithm for L_\infty regression.

**Summary Of The Review:**

In this paper, the author shows good results for an Lp regression on structured input, but the motivation towards this technique is not well presented. It also lacks a discussion on related work and a decent experimental comparison with existing techniques.

---

### Official Review · Reviewer_2tK5 · 2021-11-03

**Correctness:** 4
**Technical Novelty And Significance:** 3
**Empirical Novelty And Significance:** 2
**Recommendation:** 8
**Confidence:** 3

**Main Review:**

Here are the strengths of the paper:

(+) Theoretically, the results in the paper are solid and in particular, the result for $p=\infty$ is new and is essentially the first non-trivial sublinear time result for $L_\infty$ regression problem. I really like the $L_\infty$ regression result.

(+) The paper also presents a result for the general $A$ case (where the improvement is about a quadratic improvement).

(+) The paper is pretty well-written (some minor typos at the end of this section)

Most of my complaints are more from the more practical aspects of the results in the paper:

(-) The improvements in this paper only take effect for $d>n^{2/p}$, which for fixed $p$ is still pretty large. Of course the improvement is much more dramatic for $p=\infty$.

(-) The paper presents some motivation for $L_p$ regression: it would be useful to clarify which of those need $p>4$. In other words, what are the practical applications for the case of $p>4$ and $d>n^{2/p}$ (for the specific structured $A$ considered in this paper)? I suspect this can handled by motivating the $L_\infty$ regression problem on Vandermonde matrix.

(-) It would be better to if in the discussion of techniques in Section 1.2, there is more clarity on how these techniques differ from existing techniques in the area. E.g. in the rounding and truncating step the only "baseline" provided is the simple rounding to nearest power of $(1+\epsilon)$-- is there no "better" technical baseline to compare the techniques presented in the paper? If the rounding and truncating techniques really does not have any other baseline, then this part should probably be emphasized more.

(-) Since the contribution of the paper is more on the theoretical side, it would have been good to see more proof details in the main part of the paper.

(*) This is not really a concern but the paper did not have an ethics statement. While I can understand that the authors think that since the results in this paper are theoretical there are not ethical issues. While it is correct that there are no direct ethical issues, I would encourage the authors to think about potential mis-uses of the the algorithm by someone who potentially uses the results in this paper. Listing those in an ethical statement could be of use to others.


Detailed Comments for the authors
----------------------------------

[Pg. 2, line below Thm 1.2] Is the claim on $L_\infty$ regression for any $A$? Please clarify.

[Pg. 3, Thm 1.5+1.6] What are the best known results for these matrices prior to this work?

[Pg. 3, 3 lines below Thm 1.6] "complexity complexity"

[Pg. 6, first display equation] $\mathbf{s}$ should be $\mathbf{x}$.

Update after rebuttal
------------------------

I like the new result about query complexity of $L_p$ regression problem that the authors have added. Along with the result on $L_\infty$ for (noisy) Vandemonde matrix, I think the paper is above the accept bar, so am changing my score to a 8.

**Summary Of The Paper:**

This paper considers the problem of $L_p$-regression. I.e. given an $n\times d$ matrix $A$ and a vector $b\in\mathbb{R}^n$, compute an approximation $\tilde{x}\in\mathbb{R}^d$ such that ||A\tilde{x}-b||_p <= (1+\epsilon)\min_{x\in\mathbb{R}^d} ||Ax-b||_p. In particular, the focus of this paper is on the case of large $p$ and when the matrix $A$ has structure (specifically when A is the Vandermonde matrix or matrices that are sum of sparse and Vandermonde/low rank matrices) and the goal is to perform the task in sub/near-linear time.


This paper focuses on the well studied method of sub-sampling the rows of $A$ (and re-scaling it) to reduce the problem size where more expensive iterative methods can be applied.

For simplicity we only talk about the results for the case of A being Vandermonde matrix. The results in this paper bring down the previous best know runtime bound of $O(n\log^2{n})+d^{p/2+C}\mathrm{poly}(1/\epsilon)$ (for $p>4$ and some fixed constant $C$) to $O(n\log^2{n})+p^3d^{3/2}\mathrm{poly}(1/\epsilon,\log{n})$. Thus, for values $d >n^{2/p}$, this presents an improvement. The improvement is especially stark when we take $p=\infty$, for which case this paper presents the _first_ sublinear-time $L_\infty$ regression.

Technically the results in the paper follow from roughly three steps. The first step is to argue that for the structured matrices considered in this paper the rank of the regression problem is much smaller than the bound of $d^{\Omega(p)}$ for arbitrary $A$. This gives the desired result if $b=\mathbf{0}$ but to handle general $b$ needs more work. In particular, if all the entries of $b$ were roughly the same then the rank result is enough. Thus, it seems like the bulk of the technical work is in showing how to round and truncate $b$ and then in the third step solve the regression problem in "groups" where in each group, the sub-vector of $b$ is essentially "fixed."

There are some experimental results presented that validate the results in the paper on Vandermonde matrix as well as results for general matrices (where the paper represents a quadratic improvement over the best-known results).

**Summary Of The Review:**

Theoretically, the paper seems nice but it could be made stronger by providing motivations for the specific problem being studied in the paper. For the latter reason the paper is not a clear accept for me.

---

### Decision · Program_Chairs · 2022-01-20

**Decision:**

Accept (Poster)

**Comment:**

Dear Authors,

The paper was received nicely and discussed during the rebuttal period. There is consensus among the reviewers that the paper should be accepted:

- The new result about query complexity of regression problem that the authors have added. Along with the result on
 for (noisy) Vandemonde matrix, these make the paper lie above the accept bar.
- The authors have providing satisfying clarifications during the rebuttal that convinced reviewers to increase further their scores.

The current consensus is that the paper deserves publication.

Best AC